# A hybrid MCDM model combining Fuzzy-Delphi, AEW, BWM, and MARCOS for digital economy development comprehensive evaluation of 31 provincial level regions in China

**Haoran Zhao** [1]*, **Yuchen Wang**[2], **Sen Guo**[3]

**1** School of Economics and Management, Beijing Information Science & Technology University, Beijing, China, **2** School of Management, Dalian University of Finance and Economics, Dalian, China, **3** School of Economics and Management, North China Electric Power University, Beijing, China

* haoranzhao0118@163.com

## Abstract

With the deepening of a new round of technological revolution and industrial reform, digital technology has been continuously innovated and widely penetrated into various economic fields. The digital economy (DE) is gradually becoming the focus of China's economic development planning and a new engine to enhance national strength. Evaluating the development level of DE in various regions is conducive to timely discover the shortcomings in China's DE development, as well as provide an important basis for putting forward corresponding policy suggestions. This investigation established a hybrid multi-criteria decision making (MCDM) model for evaluating DE development of 31 provincial level regions in China ranging from 2015 to 2020. Firstly, the evaluation indicator system is established from digital infrastructure, integrated development, social benefits, innovation ability, and electronic-commerce dimensions containing 17 quantitative sub-criteria based on Fuzzy-Delphi method. Secondly, integrated weights of 17 sub-criteria from 2015 to 2020 are computed in terms of objective weights calculated by the anti-entropy weight (AEW) approach from 2015 to 2020 and subjective weights obtained via the best-worst method (BWM). Thirdly, MARCOS model is applied to evaluate the DE development degree of various regions in China ranging from 2015 to 2020. Case analysis illustrates that the DE development of Guangdong, Jiangsu, Zhejiang, and Beijing always maintain in the top four from 2015 to 2020, while the southwest and northwest regions in China are obviously fall behind others. And the DE development degree of various regions is primarily affected under the integrated development performance, innovation ability performance, and social benefits performance. Therefore, the backward regions should emphasize the development of software industry and information technology industry. The robustness of the proposed MCDM model combining Fuzzy-Delphi, AEW, BWM and MARCOS is discussed employing three similarity coefficients of rankings. And it is verified that the proposed MCDM model has superior robustness and validity in evaluating DE development.

**Data Availability Statement:** All relevant data are within the manuscript and its Supporting Information files.

**Funding:** This paper is supported by Qin Xin Talents Cultivation Program, Beijing Information Science & Technology University, under Grant No. QXTCPC202113. The funder plays a significant role in study designing and preparation of the manuscript.

**Competing interests:** The authors have declared that no competing interests exist.

# 1 Introduction

Digital economy, as a new economic operation form in the later stage of industrialization process, takes digital resources as the primary production factors and information network as a critical supporter to make efficient utilization of multiplex production resources in society via modern information communication and internet technology, so as to obtain greater economic benefits [1]. In the progression of DE development, a series of the latest information techniques, including cloud computing, big data, and artificial intelligence (AI), have gradually appeared providing technical assistance for further expanding the scale of new-type industries and economies, thus gradually changing the operation mode of the national economy, and opening a new economic era in which "Internet +" is the primary driving force for the national economic and social development [2, 3]. Therefore, the development of DE will become the primary drivers of economic growth and the propeller to enhance the comprehensive national strength in the future. In the context of global economic transformation and upgrading, aiming at finding new drivers of development and achieve steady and healthy economic development, the Chinese government takes the DE as the focus of its development plan. Since October 2010, China has emphasized the need to accelerate the integration of the internet and the substantial economy, accelerate the development of the DE, and release new space for economic development. In September 2016, the G20 Hangzhou summit released "the G20 digital economy development and cooperation initiative" and issued the "digital economy initiative". In July 2020, the National Development and Reform Commission clearly put forward to firmly grasp the golden window period of the DE development, make many industrial fields become digital test sites, and accelerate the transformation of China's industries to digitization. In October 2020, Chinese government proposed to focus on digital industrialization and industrial digitization, give impetus to the deep combination and development of DE and substantial economy, improve the international competitiveness of digital industrial clusters, and meet the arrival of the intelligent era. In 2022, Chinese government work report once again indicated the necessity of developing DE, and put forward to accelerating the digital development, building digital China and specific goals.

With the accelerating pace of China's digital transformation, the development vitality of DE is enhancing. "The White Paper on the Development of China's Digital Economy (2021)" demonstrates the DE is slightly affected by the COVID-19 epidemic situation and continues to play an important part in accelerating economic growth. In 2020, China's DE scale was 540 million dollars, ranking the second all over the world, and the growth rate ranked the first in the world with a year after year increase of 9.6%. Simultaneously, the driving force of DE on economic growth is becoming more and more obvious. In 2020, the scale of DE occupied 38.7% of GDP in China. It is demonstrated that DE is gradually becoming a new driving force for China's economic growth. However, while China's DE is developing at a high speed, it also has defects, showing the characteristics of "large but not strong" as a whole. Hence, it is particularly important to investigate the development level of China's DE and tap the development shortcomings to improve the development quality of DE [4].

In accordance with the above background, considering about the development of DE in China's provinces, this investigation will establish an index system for evaluating the DE development. Weights of sub-criteria will be determined combining subjective weights obtained by the BWM and objective weights calculated by the AEW approach. The comprehensive evaluation model will be established based on MARCOS model to estimate the development level of DE in 22 provinces, 5 autonomous areas (Inner Mongolia autonomous area, Xinjiang Uygur autonomous area, Guangxi Zhuang autonomous area, Ningxia Hui autonomous area, and Tibet autonomous area), and 4 provincial level megacities (including Beijing, Shanghai, Tianjin, and Chongqing). Taiwan, Hongkong, and Macau are excluded owing to the shortage of

related statistical data. Through empirical analysis and comparison discussion, diversities in the development level of DE among various provinces, autonomous areas, and megacities can be discovered, so that local governments can formulate targeted policies and measures to improve the development level of DE.

The primary contributions of our research are as below.

Firstly, the comprehensive evaluation indicator system is built from digital infrastructure, integrated development, social benefits, innovation ability, and electronic-commerce dimensions containing 17 sub-criteria based on Fuzzy-Delphi approach taking the knowledge and judgments of experts into consideration.

Secondly, the subjective weights are identified applying the BWM and the objective weights are computed via the AEW method. The integrated weights are obtained via the basic principle of moments estimation by calculating the subjective and objective weights coupling coefficients. Thus, the actual data information and the attributions of sub-criteria themselves can be sufficiently considered so that the reliability of sub-criteria weights can be improved.

Thirdly, a new MCDM method named MARCOS, will be employed to estimate the development level of DE in 22 provinces, 5 autonomous areas, and 4 provincial level megacities in China, which can enrich the methodology of digital economy development evaluation.

Fourthly, the robustness of the established MCDM model consisting of the Fuzzy-Delphi method, BWM, AEW method, and the MARCOS model is verified. Spearman's rank correlation coefficient ($r_S$), weighted Spearman's rank correlation coefficient ($r_W$), and WS similarity coefficient ($WS$) are employed to check the robustness of provincial level regions rankings obtained by the established MCDM model and four comparison models.

The rest sections of this investigation are designed as below. Previous literature is reviewed in Section 2. Section 3 introduces the primary methodologies applied to build the hybrid MCDM model for DE development evaluation. Section 4 describes the process of establishing the index system for comprehensive evaluation of DE development and introduces the sub-criteria contained in final index system. Case analysis is conducted in Section 5. The robustness of the established hybrid MCDM model is verified in Section 6. Section 7 draws conclusions.

## 2 Literature review

To evaluate the development level of DE, it is imperative to build a comprehensive evaluation indicator system which can accurately and comprehensively describe the characteristics of DE and elements required for DE development. In 2014, the Organization for Economic Cooperation and Development (OECD) released the "Measuring the Digital Economy-A New Perspective" report, which constructs the evaluation index system of DE development from perspectives of intelligent infrastructure, social application, innovation ability, as well as growth and employment, including 38 indicators [5]. In 2016, aiming at evaluating the performance of European DE development level and track the evolution of digital competitiveness of European Union (EU) countries, the EU issued "Digital Economy and Society Index (DESI) in 2016". The EU DE and society index system constructs the evaluation indicator system of DE development from five perspectives of network connection, human resources, network application, digital technology integration and digital public service, including more than 30 indicators. The indicator selection has a high theoretical level and certain rationality, however, a relatively simple weighted sum was used as the final result when calculating the DE development index [6]. In 2018, the U.S. Bureau of Economic Analysis issued "Defining and Measuring Digital Economy" report, which proposed that the DE is composed of three aspects containing DE infrastructure, electronic-commerce and digital media and constructed price and quantity indicators to measure DE gross production and added value according to these

three aspects. The index system, containing 12 indicators, highlights the important position of digitization in economic growth, but ignores the great value that digitization brings to government affairs, people's livelihood, environmental protection and other industries [7]. In 2017, the Digital China Research Institute of the National Information Center put forward the "five in one" index system to measure the DE in the report of "Digital China Construction Communication". This index system includes technical capacity, core development and guarantee level dimensions. And the number of primary, secondary and tertiary indicators contained in the index system are 3, 12 and 37, respectively. The index system is relatively comprehensive, but involves more qualitative indicators, of which the preference values are difficult to obtain, and does not consider the benefits of emerging technologies for the DE [8]. In 2017, Chinese Academy of Information and Communications issued the "White Paper on the Development of China's Digital Economy", which gives the evaluation index system of China's DE index composed of three parts. The first part is the leading index, which is composed of eight indicators. The second part is the consistency index, which is composed of 10 indicators. The third part is the lag index, which is composed of four indicators. Compared with the previous index system, this index system adds lagging indicators, and considers the digital economic growth brought by new digital technology, but does not consider the risk factors brought by the development of DE and the influencing factors of people's livelihood such as electronic-government and electronic-medicine [9]. In 2017, the Shanghai Academy of Social Sciences issued the "Blue Book on Digital Economy", which constructed the evaluation indicator system of global DE competitiveness from the perspectives of infrastructure, industry, innovation and governance, and analyzed the DE competitiveness of major countries in the world [10]. In September 2018, the Digital Economy Research Institution and other institutions jointly issued the "Global Digital Economy Development Index in 2018", which constructs a DE development indicator system from digital infrastructure, digital consumers, digital business ecology, digital public services, as well as digital education and scientific research perspectives, including 16 indicators. This index system not only evaluates the development level of DE, but also evaluates the industrial structure and development path of DE [11]. Zhang et al. investigates the overall development of China's DE from the perspectives of infrastructure, primary and advanced applications of information and communication technology, enterprise digital development as well as the development of information and communication technology industry. The research shows that the development of China's DE shows an upward tendency, but the increase rate shows a slowing trend. However, the index system greatly highlighted the economic benefits and lacks the measurement of social benefits, and does not consider the impact of network security level on the development of DE [12].

For methods employed in measuring the DE development, Chen comprehensively used entropy method, Dagum-Gini coefficient method and panel Tobit model to explore the development level, regional diversities and driving factors of DE in various provinces and cities utilizing the panel data of 30 provinces and cities ranging from 2013 to 2019. Results showed that from the perspective of driving factors, the level of economic development, industrial structure, human capital and scientific and technical development could strongly promote the development of DE, while government intervention will inhibit the development of DE. Among them, scientific and technical development and intellectual property protection have a great influence on the DE development [13]. Li and Han constructed the quantitative indicator system of DE development level from digital infrastructure, digital industrialization and industrial digitization combined with the connotation and extension of DE. Then the development tendency of DE in China from 2010 to 2018 was measured based on entropy method, and a grey prediction approach was constructed to forecast the development tendency of DE from 2019 to 2028 [14]. Dong constructed the index system to evaluate the DE of China from digital

infrastructure, integrated development, innovation ability, social benefits and electronic-commerce. Then the scores of DE development level of all provinces in China from 2015 to 2019 were calculated based on PCA-EM model and analyzed the regional development differences and the causes of the differences with the help of coefficient of variation analysis and geographic detectors [15]. Li employed DEA model to comprehensively evaluate and analyze the development efficiency of China's DE from the static and dynamic aspects utilizing the constructed evaluation indicator system of DE development efficiency. Then, in accordance with relevant economic theories, the basic assumptions affecting the development efficiency of DE are put forward, and an empirical test is conducted by using Tobit model [16]. Li constructed a comprehensive, scientific, and complete comprehensive evaluation indicator system of DE development level from the perspectives of basic resources, integrated development, innovation ability, social benefits, and network security. Then, taking Jiangsu Province as an example, the DE development level of 13 cities in Jiangsu Province from 2014 to 2017 was calculated by using entropy weight approach and approximate ideal solution ranking model and a comparative analysis was also conducted on the overall level and five dimensions of DE development in various cities [17]. Chen et al. employed nighttime light remote sensing data to estimate the DE and utilized the Zipf's law to evaluate the DE growth from the city level [18]. Nela Milošević et al. employed the Composite I-distance Indicator (CIDI) methodology to evaluate as well as rank digital performance of 28 countries in European Union [19].

On the basis of the analysis of existing literature, it can be found that firstly foreign research on DE started earlier, and most of the index systems have been tested for a long time with high credibility and authority. Domestic research on DE started relatively late, but the evaluation indicator system of DE development is more comprehensive. Secondly, it is difficult to find literatures on the comprehensive evaluation of DE development employing MCDM model, while traditional researches on the measurement of DE development primarily depended on the statistical data [20, 21]. Nevertheless, the coherence and comparable objective data are performed infrequently and hard to be collected, as it needs a great amount of labor work and not all the objective data are usable for various areas. Owing to the data restriction, the development of DE is hard to be analyzed. Thirdly, most of the previous investigations are on the basis of the national level, which would neglect the differences between regions. Considering that all provincial level regions in China have their own DE development strategies and plans, evaluating the DE development level of each provincial level region can provide more policy support for the DE development of the relevant provincial level region. Therefore, this paper evaluates and analyzes the development level of provincial DE in China, and puts forward targeted suggestions combined with the development characteristics of provincial DE, so as to accelerate the high-level and high-quality development of China's DE.

The index system for evaluating the DE development is established based on Fuzzy-Delphi considering about experts knowledge and experience. And the BWM is employed to determine the subjective weights, as it only needs to compare the importance of each sub-criterion with the best and the worst sub-criteria, which is time-saving and convenient based on experts' judgments. And the AEW method is utilized to determine the objective weights based on objective data. These two methods have been employed in many fields to determine sub-criteria weights in comprehensive evaluation, such as aircraft selection [22], green supplier selection [23], and enterprise debt decision [24]. To comprehensively considering the importance of experts judgments and objective data, a weight integrating method is employed to combining subjective weights and objective weights based on the basic principle of moments estimation. Then integrated weights can be obtained. Afterwards, a new MCDM method named MARCOS, proposed by Željko Steviã and Dragan Pamučar in 2020, will be employed to estimate the development level of DE in 22 provinces, 5 autonomous areas, and 4 provincial level

megacities in China. MARCOS model can take the positive and negative ideal solutions into account at the same time, and rank the provincial level regions based on the utility functions, which can make the results have superior robustness and accuracy [25, 26]. Therefore, a MCDM framework combining Fuzzy-Delphi, the BWM, the AEW, and MARCOS methods is established for DE development comprehensive evaluation of 31 provincial level regions in China.

## 3 Methods

The hybrid MCDM model for comprehensive evaluation of DE development is systematic and rigorous integrating the Fuzzy-Delphi method, BWM, AEW method and MARCOS model. The concrete process is illustrated as Fig 1.

### 3.1 Fuzzy-Delphi method

Delphi method has been extensively applied in decision making researches, which can be used to collect the most valid opinions from experts [27]. In traditional Delphi method, experts can receive feedback and reconsider their opinions provided previously via four rounds consultation [28], of which the process is superfluous and hard to gather consistent judgments via several rounds consulting. Hence, to deal with the above demerits, the Fuzzy-Delphi method, combining traditional Delphi method and fuzzy theory, has been established. In this method, experts should make three points judgment using triangular fuzzy numbers (TFNs) [29]. Membership degree functions (MDFs) are utilized to indicate the judgments of experts. Advantages of MDFs are that experts do not need to adjust their judgments through several rounds consultation and MDFs can sufficiently use all judgments. Due to the superiority of the Fuzzy-Delphi, many researches utilized it to construct the evaluation indicator system [30]. The steps of Fuzzy-Delphi method in detail are described as below.

**Step 1**: By gathering the judgments of experts, we can identify the upper and bottom bounds of confidence interval [0, 10] for each indicator. If the value is larger, the indicator is more critical. The upper bound represents the most positive value, while the bottom one implies the most conservative value.

**Step 2**: Calculating the conservative TFN $(C_L^i, C_M^i, C_U^i)$ and positive TFN $(P_L^i, P_M^i, P_U^i)$ of each indicator, of which $C_L^i$ and $P_L^i$ are the smallest values among the most conservative and positive values considering about experts opinions, $C_M^i$ and $P_M^i$ are the geometric average values of the most conservative and positive values, and $C_U^i$ and $P_U^i$ are the largest values of the most conservative and positive values.

**Step 3**: Examining the consistency of experts' opinions and identifying the coherent value $G_i$ of indicator $i$ [30, 31].

(1) If $C_U^i < P_L^i$, the judgment for indicator $i$ are coherent, and $G_i$ is determined by:

$$G_i = \frac{C_M^i + P_M^i}{2} \tag{1}$$

(2) If $C_U^i > P_L^i$, if the gray scope value $Z^i = C_U^i - P_L^i$ is smaller than $M^i = P_U^i - C_M^i$, $G_i$ is calculated by:

$$G_i = \frac{[(C_U^i \times P_M^i) - (P_L^i \times C_M^i)]}{[(C_U^i - C_M^i) + (P_M^i - P_L^i)]} \tag{2}$$

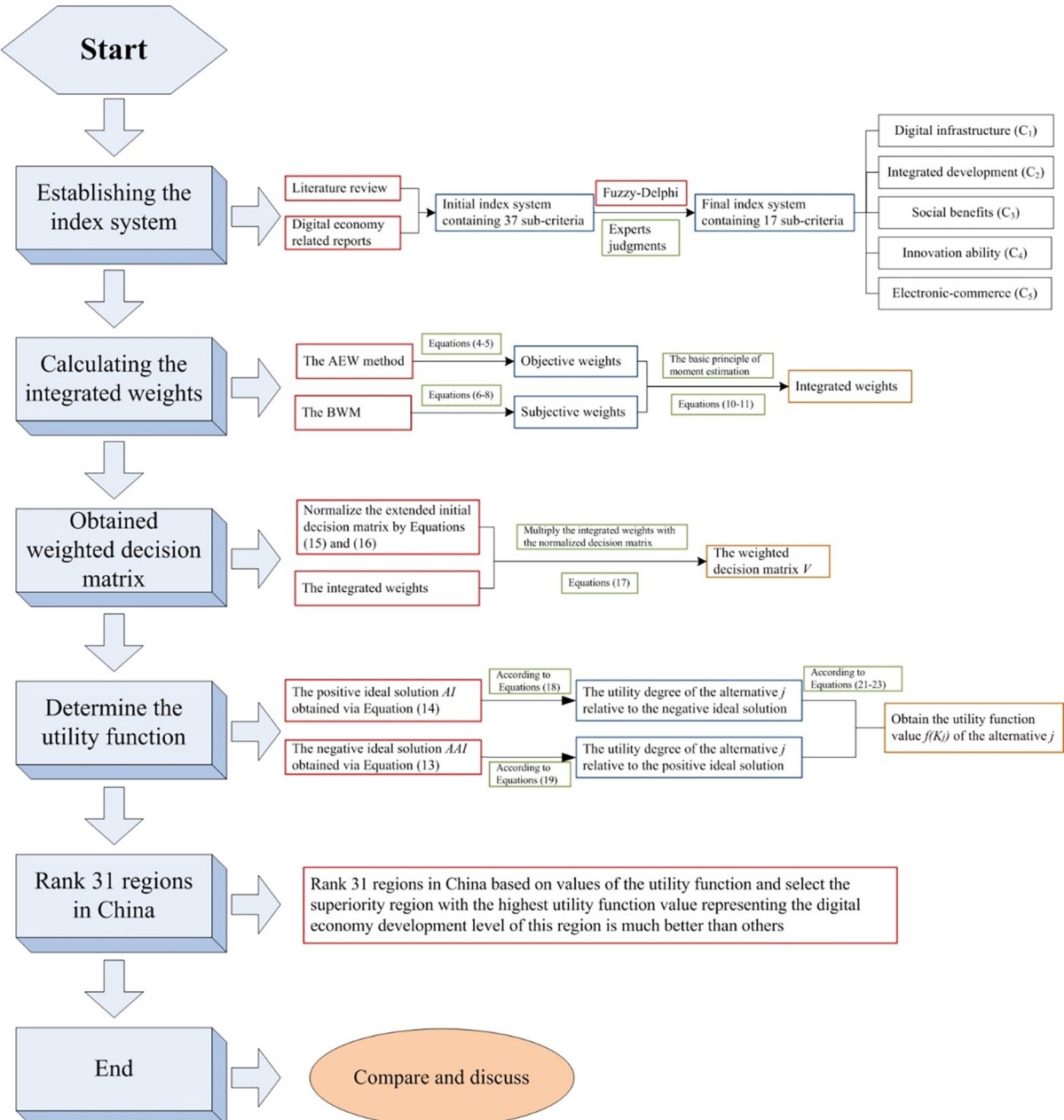

**Fig 1. The concrete procedure of the hybrid MCDM model for comprehensive evaluation of DE development.**

While if the grey scope value $Z^i = C^i_U - P^i_L$ is larger than $M^i = P^i_U - C^i_M$, the opinions of experts will be conflicting. Then we should process the Steps 1 and 2 again until the opinions are coincident. $G_i$ demonstrates the consistent degree identified by experts judging the important degree of each indicator. The larger the value of $G_i$, the more important the indicator will be. To find an optimal value to verify the important degree of each indicator, the geometric mean value of the upper and bottom bounds of all indicators is calculated, deeming as the confident value to screen final indicators.

## 3.2 Sub-criteria weighting method on the basis of AEW and BWM

(1) AEW method

Supposing that there exist $m$ probable situations in the system, and every situation occurs with a probability of $p_j$ ($j$ = 1,2,...,m), then the entropy is calculated as [32]:

$$h = -\sum\nolimits_{j=1}^{m} p_j \ln p_j \tag{3}$$

where $0 \leq p_j \leq 1$, and $\Sigma_j p_j = 1$.

For AEW approach, assuming the amount of alternatives is $m$ and the amount of sub-criteria is $n$, the value of sub-criteria $i$ with regard to alternative $j$ is $x_{ij}$ ($i$ = 1,2,...,n, $j$ = 1,2,...,m), hence, the assessment matrix is described as X = $[x_{ij}]_{n \times m}$. Thus, the anti-entropy value of every sub-criterion can be obtained via [33]:

$$h_i = -\sum\nolimits_{j=1}^{m} r_{ij} \ln(1 - r_{ij}) \tag{4}$$

where $r_{ij} = x_{ij}/\Sigma_j x_{ij}$. Via standardizing the anti-entropy value, the objective weight $w_{1i}$ of every sub-criterion is calculated by:

$$w_{1i} = h_i / \sum\nolimits_i h_i \tag{5}$$

(2) The BWM

The BWM is a pair comparison-based approach, the fundamental theory of which is similar with AHP, but simpler than that. AHP method determines weights via pair-wise comparisons after $\frac{n(n-1)}{2}$ comparisons (supposing that there are $n$ sub-criteria). Nevertheless, the BWM only needs *2n-3* comparisons, *1* for comparing the best sub-criterion with the worst one, *n-2* for comparing the best sub-criterion with others, and *n-2* for comparing the others with the worst one [34, 35]. The concrete process of the BWM is introduced as below.

**Step 1**: Choose the best and the worst sub-criteria from the constructed final sub-criterion system {$C_1$, $C_2$,...,$C_n$}. The best sub-criterion is the most critical and desirable one which can accurately describe the distinguishing features of the alternatives, nevertheless, the worst one is contrary.

**Step 2**: Compare the best one with others using numbers from 1 to 9. The larger value indicates the best one is much more crucial than others. Comparing results are written as:

$$A_B = (a_{B1}, a_{B2}, \ldots, a_{Bn}) \tag{6}$$

where $a_{Bi}$ implies the critical degree of the best one $B$ over indicator $i$, and $a_{BB} = 1$.

**Step 3**: Compare other indicators with the worst one using numbers from 1 to 9. Results are:

$$A_W = (a_{1W}, a_{2W}, \ldots, a_{nW})^T \tag{7}$$

where $a_{iW}$ represents the importance degree of indicator $i$ over the worst one $W$, and $a_{WW} = 1$.

**Step 4**: Compute the optimal weights $(w_1^*, w_2^*, \ldots, w_n^*)$ of all indicators. To obtain the optimal weight of every sub-criterion, we should minimize the maximum differences $\{|w_B - a_{Bi}w_i|, |w_i - a_{iW}w_W|\}$ for all sub-criteria, and the formula is linearized referred to Ref

**Table 1. Consistent Indicator (CI).**

| $a_{BW}$ | 1 | 2 | 3 | 4 | 5 | 6 | 7 | 8 | 9 |
|---|---|---|---|---|---|---|---|---|---|
| CI | 0.00 | 0.44 | 1.00 | 1.63 | 2.30 | 3.00 | 3.73 | 4.47 | 5.23 |

[36]:

$$\min \quad \varepsilon$$

$$\text{s.t.}$$

$$|w_B - a_{Bi}w_i| \leq \varepsilon, \quad \text{for all } i$$

$$|w_i - a_{iw}w_w| \leq \varepsilon, \quad \text{for all } i \tag{8}$$

$$\sum_i w_i = 1,$$

$$w_i \geq 0, \quad \text{for all } i$$

Afterwards, the optimal weights $(w_1^*, w_2^*, \ldots, w_n^*)$ are determined in terms of the above formula. Hence, subjective weights written as $w_{2i}$ of sub-criterion $i$ are identified.

**Step 5**: Testify the comparing consistence by coherence test. The greatest value of $\varepsilon$ is calculated through processing the Formula (8). Table 1 illustrates the coherent indicator which are selected based on the values of $a_{BW}$ ranging from 1 to 9. Hence, the consistent ratio can be determined via:

$$\text{Coherent ratio} = \frac{\varepsilon^*}{\text{Coherent indicator}} \tag{9}$$

The smaller the consistent ratio is, the more consistent the comparing will be.

(3) Integrated weights coupling subjective weights and objective weights

The BWM, applied to determine subjective weights, makes a significance ranking of sub-criteria in terms of experts' knowledge and experience. However, it can only reflect the subjective opinion of experts about the critical degree of sub-criteria. The AEW method, applied to calculate objective weights, makes a significance ranking of sub-criteria in terms of the objective data of each sub-criteria in regard to alternatives. However, it is difficult for subjective weights to reflect actual data information of sub-criteria, while objective weights can reflect actual data information but will change with actual data change of alternatives. Thus, the volatility of objective weights is stronger than subjective ones. Taking the merits and demerits of subjective and objective weights into consideration, recent researches focused on weights combination methods [37–39]. Integrated methods combining subjective and objective weights cannot only avoid the issue that objective weights emphasize actual data and ignore the sub-criteria attributes bringing about the unreasonable weights results, but also avoid the issue that experts' opinions are too subjective, so as to obtain more effective weights.

In this investigation, integrated weights of subjective and objective weights are estimated in terms of the basic principle of moment estimation. Owing to the difference of the relative significance of subjective and objective weights for sub-criteria, subjective and objective weight coupling coefficients are obtained via:

$$\begin{cases} \delta_i = \dfrac{w_{1i}}{w_{1i} + w_{2i}} \\ \varepsilon_i = \dfrac{w_{2i}}{w_{1i} + w_{2i}} \end{cases} \tag{10}$$

where $\delta_i$ and $\varepsilon_i$ are the objective and subjective weight coupling coefficients of sub-criterion $i$, respectively.

Afterwards, the integrated weight $\theta_i$ of sub-criterion $i$ integrating subjective and objective weights of sub-criterion $i$ is computed via:

$$\theta_i = \frac{\delta_i w_{1i} + \varepsilon_i w_{2i}}{\sum_{i=1}^{n} \delta_i w_{1i} + \varepsilon_i w_{2i}} \tag{11}$$

## 3.3 Comprehensive evaluation model based on MARCOS

Measurement of alternatives and ranking in accordance with compromise solution (MARCOS), proposed by Željko Steviā and Dragan Pamučar in 2020 [40], will be applied to estimate the development of the DE of 31 provinces in China. In MARCOS method, through comparing the reference values of alternatives with ideal values, the utility functions of alternatives are determined, and the compromise ranking related to the ideal alternative and the negative ideal alternative is attained. MARCOS method can consider positive and negative ideal solutions simultaneously, and consider the probability of a great amount of standards and alternatives while keeping the steadiness of the model, so as to obtain the results with good robustness and precision. The concrete procedure is expressed as below.

**Step 1**: Establishing extended original matrix $X = [x_{ij}]_{n \times m}$. The positive ideal solution ($AI$) and negative ideal solution ($AAI$) are defined to expand the matrix $X = [x_{ij}]_{n \times m}$. The $AI$ is the alternative with the best characteristics, while the $AAI$ is contrary.

$$X = \begin{array}{c} \\ AAI \\ A_1 \\ A_2 \\ \cdots \\ A_n \\ AI \end{array} \begin{pmatrix} C_1 & C_2 & \cdots & C_m \\ x_{aa1} & x_{aa2} & \cdots & x_{aam} \\ x_{11} & x_{12} & \cdots & x_{1m} \\ x_{21} & x_{22} & \cdots & x_{2m} \\ \cdots & \cdots & \cdots & \cdots \\ x_{n1} & x_{n2} & \cdots & x_{nm} \\ x_{ai1} & x_{ai2} & \cdots & x_{aim} \end{pmatrix} \tag{12}$$

Among which, in accordance with the attribute of the sub-criterion, $AAI$ and $AI$ are determined by

$$AAI = \min_{j} x_{ij}, \text{if } i \in B \text{ and } \max_{j} x_{ij}, \text{if } i \in C \tag{13}$$

$$AI = \max_{j} x_{ij}, \text{if } i \in B \text{ and } \min_{j} x_{ij}, \text{if } i \in C \tag{14}$$

where $B$ represents maximum type sub-criteria and $C$ indicates minimum type sub-criteria.

**Step 2**: Standardizing extended initial matrix $N = [n_{ij}]_{n \times m}$. The standardized decision matrix is attained by normalizing the matrix $X = [x_{ij}]_{n \times m}$.

$$n_{ij} = \frac{x_{ij}}{x_{aj}}, \text{if } i \in B \tag{15}$$

$$n_{ij} = \frac{x_{aj}}{x_{ij}}, \text{if } i \in C \tag{16}$$

where $x_{ij}$ and $x_{aj}$ are the elements in initial matrix $X = [x_{ij}]_{n \times m}$.

**Step 3**: Determining weighted normalized decision matrix $V = [v_{ij}]_{n \times m}$ via:

$$v_{ij} = n_{ij} \times \theta_i \tag{17}$$

**Step 4**: Calculating the utility degree $K_j$ of alternative $j$. The utility degrees $K_j$ relative to the negative and positive ideal solutions are computed via the following formula:

$$K_j^- = \frac{S_j}{S_{aaj}} \tag{18}$$

$$K_j^+ = \frac{S_j}{S_{aj}} \tag{19}$$

where $S(j = 1,2,\ldots m)$ represents the sum of elements in the weighting matrix $V$.

$$S_j = \sum_{j=1}^{m} v_{ij} \tag{20}$$

**Step 5**: Identify the utility function $f(K_j)$ of the alternative $j$. The concrete expression is:

$$f(K_j) = \frac{K_j^+ + K_j^-}{1 + \frac{1-f(K_j^+)}{f(K_j^+)} + \frac{1-f(K_j^-)}{f(K_j^-)}} \tag{21}$$

where $f(K_j^-)$ and $f(K_j^+)$ imply the utility functions relative to the negative and positive ideal solutions. They can be expressed as below.

$$f\left(K_j^+\right) = \frac{K_j^-}{K_j^+ + K_j^-} \tag{22}$$

$$f\left(K_j^-\right) = \frac{K_j^+}{K_j^+ + K_j^-} \tag{23}$$

**Step 6**: Ranking alternatives in accordance with the final value of the utility function. The ideal alternative is the one with the greatest utility function value.

## 3.4 The procedure of the established hybrid MCDM model for comprehensive evaluation of DE development

The detailed procedure employing the hybrid MCDM model for comprehensive evaluation of DE development of 31 provincial level regions in China is introduced as below and Fig 1.

**Step 1**: Screen the crucial indicators to construct the final comprehensive evaluation indicator system employing Fuzzy-Delphi method. Firstly, we need to collect and summarize initial sub-criteria from related reports and researches on DE development. Secondly, 5 authoritative experts in DE development research field are invited to compose the decision group. Thirdly, experts should judge the important degree of each indicator using the Fuzzy-Delphi method so that we can screen the critical sub-criteria in terms of Eqs (1) and (2). Afterwards, the final indicator system for comprehensive evaluation of DE development can be constructed.

**Step 2**: Collect the objective data for all sub-criteria to obtain the decision matrix $X = [x_{ij}]_{n \times m}$. Considering that all sub-criteria contained in final index system are objective sub-criteria, so that actual data can be collected from statistical yearbook of each provincial level region and the official website of National Bureau of Statistics. Afterwards, the decision matrix $X = [x_{ij}]_{n \times m}$ are obtained.

**Step 3**: Compute the comprehensive weights of all indicators coupling the subjective weights obtained via the BWM and objective weights obtained via the AEW. Firstly, objective weights are calculated on the basis of the decision matrix utilizing the AEW method through Eqs (4) and (5). Secondly, subjective weights are judged via the BWM taking experts' judgments into consideration through Eqs (6)–(8). Finally, comprehensive weights integrating objective and subjective weights are obtained through Eqs (10)–(11).

**Step 4**: Obtain weighted normalized decision matrix $V = [v_{ij}]_{n\times m}$. Firstly, extended initial matrix should be established combining the *AAI* and *AI* data series which can be attained through Eqs (13) and (14). Secondly, the standardized extended initial matrix $N = [n_{ij}]_{n\times m}$ can be obtained through Eqs (15) and (16). Finally, the weighted standardized decision matrix $V = [v_{ij}]_{n\times m}$ can be calculated via Eq (17).

**Step 5**: Determine the utility function $f(K_j)$ of the provincial level region $j$. Firstly, we need to compute the utility degrees of the provincial level regions relative to the negative and positive ideal solutions of the provincial level region $j$ using Eqs (18) and (19). Then the utility function $f(K_j)$ of the provincial level region $j$ can be determined via Equs (21)-(23).

**Step 6**: Rank provincial level regions in accordance with the final value of the utility function $f(K_j)$ and select the optimal provincial level region with the greatest utility function value, which represents the DE development level of this provincial level region is superior than others.

## 4 Indicator system of comprehensive evaluation on DE development

To comprehensively and accurately evaluate the development of DE of various provinces in China, the construction of comprehensive evaluation index system plays a significant role. Firstly, the initial indicator system is built from digital infrastructure, integrated development, social benefits, innovation ability, and electronic-commerce dimensions containing 31 initial sub-criteria through referring to related reports and published literature. Secondly, we utilized Fuzzy-Delphi to select the crucial indicators and construct the final index system considering 5 experts judgments, who are invited from universities, governments, and DE research institute.

During the process of screening the critical sub-criteria, experts firstly should judge the important degree of each initial indicator and the most positive and the most conservative values of each indicator are determined. Secondly, the conservative TFN ($C_L^i, C_M^i, C_U^i$) and positive TFN ($P_L^i, P_M^i, P_U^i$) of each sub-criterion are computed. Thirdly, the coherence of experts' judgments can be assessed via computing the values of $G_i$ using Eqs (1) and (2). Afterwards, the critical sub-criteria are screened via comparing values of $G_i$ with the confident value (which is 7.26 computed by the geometric average value of upper and bottom bounds of all original indicators). Results for the process of indicators screening are illustrated in Table 2, and 17 critical indicators are chosen to compose the final indicator system for DE development evaluation of 31 provincial level regions in China.

The final index system for DE development evaluation is illustrated in Fig 2, which is composed of 17 sub-criteria attributed to five aspects of digital infrastructure, integrated development, social benefits, innovation ability, and electronic-commerce. All the final sub-criteria are all qualitative and benefit-type.

For digital infrastructure, DE infrastructure is a crucial part to ensure the steady operation and development of DE. All activities of DE are upon infrastructure construction. The construction of DE infrastructure is reflected in the popularity of information techniques hardware and internet. Hence, based on experts' judgments, telephone penetration ratio, the

**Table 2. Results of Fuzzy-Delphi method in screening critical sub-criteria.**

| Perspectives | Original sub-criteria | Conservative value | | | Optimistic value | | | Mi-Zi | Consistent value |
|---|---|---|---|---|---|---|---|---|---|
| | | $C_L^i$ | $C_M^i$ | $C_U^i$ | $P_L^i$ | $P_M^i$ | $P_U^i$ | | Gi |
| Digital infrastructure | Telephone penetration ratio | 6 | 7.34 | 8 | 7 | 9.31 | 10 | 1.66 | 7.78 |
| | Internet penetration ratio | 3 | 3.34 | 5 | 4 | 5.19 | 6 | 1.66 | 4.42 |
| | The number of internet access ports | 2 | 3.78 | 4 | 4 | 4.31 | 5 | 1.22 | 4.05 |
| | The number of mobile phone base station per square kilometer | 2 | 2.31 | 3 | 3 | 3.87 | 5 | 2.69 | 3.09 |
| | Proportion of digital TV users | 3 | 3.31 | 4 | 5 | 5.35 | 6 | 3.69 | 4.33 |
| | The number of computers users per 100 people | 3 | 3.41 | 4 | 5 | 5.46 | 6 | 3.59 | 4.44 |
| | The number of internet broadband access users | 6 | 7.68 | 8 | 7 | 8.61 | 9 | 0.32 | 7.83 |
| | Long distance optical cable line length | 6 | 7.24 | 8 | 7 | 8.67 | 10 | 1.76 | 7.69 |
| Integrated development | The number of enterprises assessed by integration of industrialization and informatization | 2 | 2.57 | 3 | 4 | 4.64 | 5 | 3.43 | 3.61 |
| | Proportion of investment in information, computer and software industry | 3 | 3.86 | 5 | 4 | 5.11 | 6 | 1.14 | 4.49 |
| | The number of websites per 100 enterprises owned | 6 | 7.88 | 8 | 7 | 9.31 | 10 | 1.12 | 7.95 |
| | Software business income | 6 | 7.56 | 8 | 7 | 8.98 | 10 | 1.44 | 7.82 |
| | Express business income | 6 | 7.35 | 8 | 7 | 9.36 | 10 | 1.65 | 7.78 |
| | Number of postal outlets | 3 | 3.54 | 4 | 5 | 5.38 | 6 | 3.46 | 4.46 |
| | Total post and telecommunications business | 6 | 6.46 | 7 | 8 | 9.25 | 10 | 4.54 | 7.86 |
| | Income from information technology services | 6 | 7.21 | 9 | 8 | 9.33 | 10 | 1.79 | 8.43 |
| Social benefits | Express quantity | 6 | 7.23 | 8 | 7 | 9.07 | 10 | 1.77 | 7.73 |
| | Proportion of information industry employment | 3 | 3.21 | 4 | 4 | 4.37 | 5 | 1.79 | 3.79 |
| | Average wage of urban employees in information transmission, computer services and software industries | 6 | 7.42 | 8 | 7 | 9.46 | 10 | 1.58 | 7.81 |
| | Employment of urban units in information transmission, software and information technology services | 6 | 7.21 | 8 | 7 | 9.48 | 10 | 1.79 | 7.76 |
| | Amount of government public information | 3 | 3.32 | 4 | 4 | 4.37 | 5 | 1.68 | 3.85 |
| Innovation ability | Proportion of government expenditure on science and technology | 3 | 3.47 | 5 | 4 | 4.32 | 7 | 2.53 | 4.17 |
| | The number of patent applications of industrial enterprises above designated size | 6 | 7.68 | 8 | 7 | 8.61 | 9 | 0.32 | 7.83 |
| | Research & development funds for industrial enterprises above designated size | 6 | 7.34 | 8 | 7 | 9.21 | 10 | 1.66 | 7.77 |
| | Proportion of patent applications in electronic communication manufacturing industry | 2 | 2.58 | 3 | 4 | 4.11 | 5 | 3.42 | 3.35 |
| | Proportion of research & development funds in profits of enterprises above designated size | 2 | 2.79 | 3 | 3 | 3.87 | 5 | 2.21 | 3.33 |
| | The number of research & development personnel in enterprises above designated size | 6 | 7.61 | 9 | 8 | 9.67 | 10 | 1.39 | 8.55 |
| | Patent application density | 3 | 3.24 | 4 | 5 | 5.27 | 6 | 3.76 | 4.26 |
| Electronic-commerce | Electronic-commerce sales amount | 5 | 7.29 | 8 | 7 | 9.31 | 10 | 1.71 | 7.76 |
| | Electronic-commerce purchase amount | 6 | 6.89 | 8 | 7 | 9.26 | 10 | 2.11 | 7.67 |
| | The proportion of enterprises with electronic-commerce transactions | 6 | 6.88 | 8 | 7 | 8.98 | 10 | 2.12 | 7.64 |

number of internet broadband access users, and long-distance optical cable line length are selected. Telephone penetration ratio refers to the popularity of telephones in a region, measured by the value of all telephones in 100 people. It can reflect the degree of mobile network construction and the application of mobile network in the region. The number of internet broadband access users indicates the coverage of broadband network, which can reflect the construction of network from the perspective of broadband. The length of long-distance optical cable line can reflect the coverage of long-distance optical cable line in a certain area and the application of new internet technology.

For integrated development, the development of DE is not independent, but complementary to the traditional economy. With the help of information technology, DE innovates the

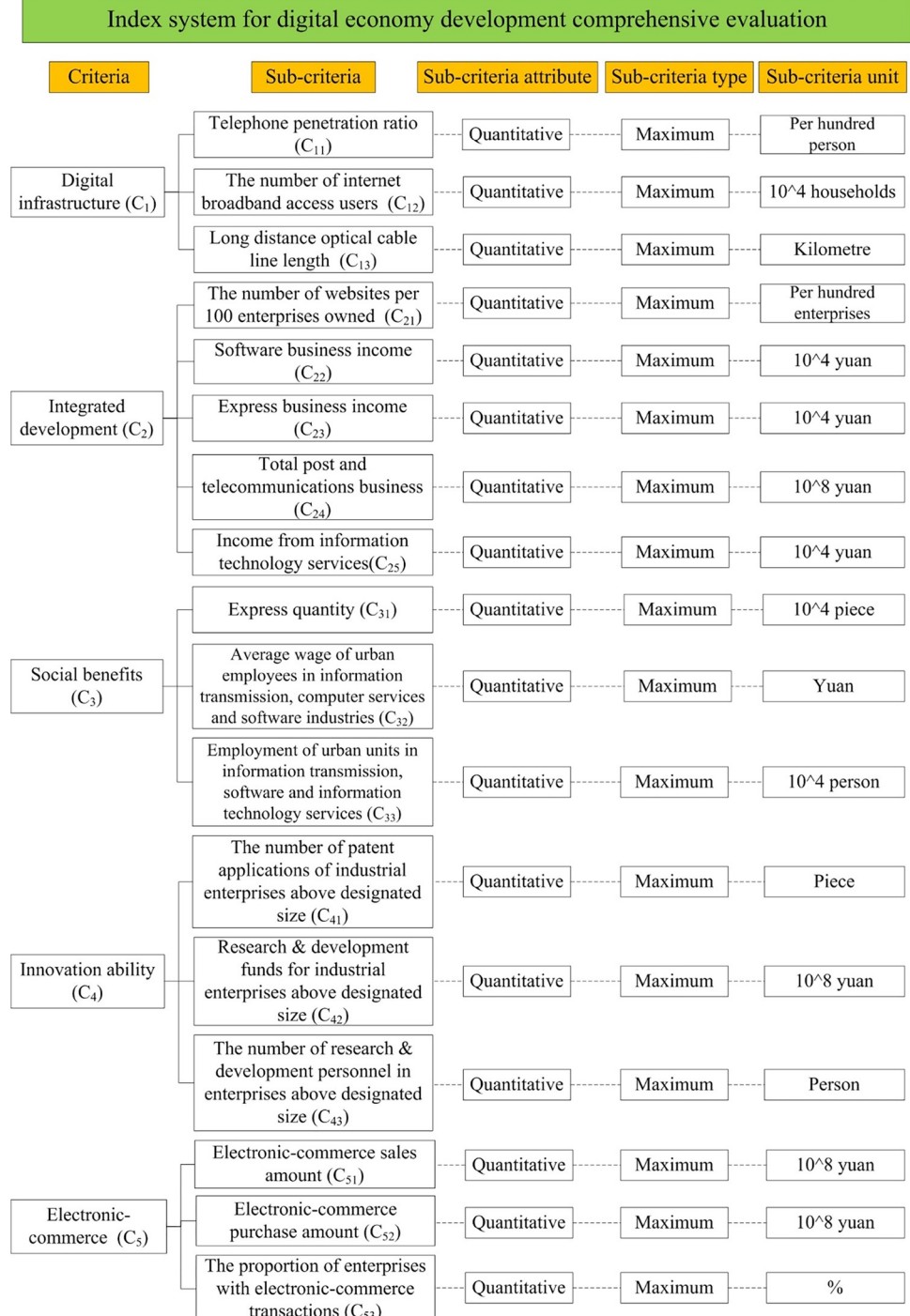

**Fig 2. Final indicator system for DE development evaluation.**

production technology and operation mode of traditional economy. In the meantime, DE also excavates more effective information for traditional economic system, so as to make more effective decisions. Therefore, this paper selects the number of websites per 100 enterprises owned, software business income, express business income, total post and telecommunications business, and income from information technology services as secondary indicators to

estimate the integrated development of DE. The number of websites per 100 enterprises owned refers to the number of websites owned by every 100 enterprises, reflecting the integration degree of the general traditional economy with the internet as the medium. Software business income and income from information technology services reflect the integration development of traditional economy and informatization from the aspects of software business and information technology services. Express business income and total post and telecommunications business reflect the closeness of DE and postal transportation industry, as the impact of DE goes deep into the postal transportation industry and the integrated development of DE is inseparable from the coordinated driving of the postal transportation industry.

For social benefits, it refers to the rational distribution of limited resources to satisfy the needs of society and people's life. Considering about experts' opinions, express quantity, average wage of urban employees in information transmission, computer services and software industries, and employment of urban units in information transmission, software and information technology services are selected as the secondary indicators. Express quantity reflects the situation of internet shopping and the impact of the development of DE on consumers' consumption mode. Average wage of urban employees in information transmission, computer services and software industries can reflect the wage level of employees in the information industry. Since the information industry is a representative part of the operation system of DE, employment of urban units in information transmission, software and information technology services can reflect the contribution of the development of DE to social employment to a certain extent.

For innovation ability, it can innovate the existing science and technology, so as to alter the mode of production and methods, achieving the purpose of industrial transformation and upgrading. The development of DE is supported by high-tech digital technologies, such as AI, cloud computing, and others, and the birth of high-technologies comes from development and innovation. The ability of independent innovation is the power source of the development of DE. The number of patent applications of industrial enterprises above designated size, research & development funds for industrial enterprises above designated size, and the number of research & development personnel in enterprises above designated size are selected as the secondary indicators to represent the innovation ability for DE development, which can reflect the importance and practice of independent innovation from the perspective of enterprises.

For electronic-commerce, it is a new product of the development of internet technology which turns the original offline trade into online trading activities through the internet. Electronic-commerce not only provides convenience for production and life, but also shows the development trend of DE. Electronic-commerce sales amount, electronic-commerce purchase amount, and the proportion of enterprises with electronic-commerce transactions are screened as the secondary indicators. Electronic-commerce sales amount refers to all the remuneration obtained by enterprises through various transactions in the network environment reflecting the output of the development of DE. Electronic-commerce purchase amount refers to the amount required by enterprises to purchase relevant goods during electronic-commerce business, which is one of the important indicators reflecting the operation scale of DE. The proportion of enterprises with electronic-commerce transactions refers to the proportion of enterprises with electronic-commerce activities in all enterprises which can reflect the scale of the development of DE.

## 5 Case analysis

After screening the critical sub-criteria employing the Fuzzy-Delphi method, the evaluation indicator system for DE development can be established. Considering that all sub-criteria are

quantitative, actual data of all sub-criteria with regard to 31regions in China can be gathered from the website of National Bureau of Statistics (http://www.stats.gov.cn/). Aiming at comparatively researching on the development level of DE in 31 regions in different years, data from 2015 to 2020 are collected, which are listed in S1 Data. Based on the collected data, case analysis can be processed.

## 5.1 Determine the integrated weights of all sub-criteria

(1) Determine the objective weights employing AEW method

Based on the collected actual data of all indicators with regard to 31 regions in China from 2015 to 2020, objective weights of all sub-criteria from 2015 to 2020 are calculated via Eqs (4) and (5). Considering that the actual data are various in different years, so that the objective weights are various in various years. Results of objective weights from 2015 to 2020 are listed in Table 3.

(2) Determine the subjective weights employing the BWM

Subjective weights are obtained via the BWM taking the opinions and judgments given by 5 selected experts. Since subjective weights only considers experts' opinions considering about their knowledge and experience without taking the actual data of indicators into account with regard to provincial level regions, hence, final subjective weights has only one set of data regardless of years. Firstly, the best and the worst indicators are chosen from the final constructed indicator system. The best indicator is the one with the great significance to describe the particular characteristics of DE development, while the worst one is contrary. After collecting the judgments of 5 experts, the best and the worst indicators chosen by 5 experts are listed in Table 4.

Secondly, 5 experts should compare the important degree of the best indicator to others and others to the worst indicator using a score ranging in the interval of [1, 9]. The value of 1

**Table 3. Objective weights from 2015 to 2020 obtained via AEW method.**

| Criteria | Sub-criteria | 2015 | 2016 | 2017 | 2018 | 2019 | 2020 |
|---|---|---|---|---|---|---|---|
| Digital infrastructure | $C_{11}$ | 0.0264 | 0.0254 | 0.0246 | 0.0242 | 0.0239 | 0.0236 |
| | $C_{12}$ | 0.0406 | 0.0391 | 0.0378 | 0.0357 | 0.0344 | 0.0339 |
| | $C_{13}$ | 0.0317 | 0.0311 | 0.0305 | 0.0302 | 0.0335 | 0.0330 |
| Integrated development | $C_{21}$ | 0.0256 | 0.0250 | 0.0245 | 0.0241 | 0.0240 | 0.0238 |
| | $C_{22}$ | 0.0817 | 0.0817 | 0.0810 | 0.0788 | 0.0768 | 0.0829 |
| | $C_{23}$ | 0.0964 | 0.0940 | 0.0946 | 0.0911 | 0.0944 | 0.0948 |
| | $C_{24}$ | 0.0469 | 0.0519 | 0.0475 | 0.0411 | 0.0387 | 0.0384 |
| | $C_{25}$ | 0.0737 | 0.0757 | 0.0760 | 0.0808 | 0.0798 | 0.0862 |
| Social benefits | $C_{31}$ | 0.1034 | 0.1004 | 0.1031 | 0.1019 | 0.1068 | 0.1080 |
| | $C_{32}$ | 0.0276 | 0.0270 | 0.0265 | 0.0261 | 0.0261 | 0.0262 |
| | $C_{33}$ | 0.0616 | 0.0610 | 0.0632 | 0.0663 | 0.0640 | 0.0654 |
| Innovation ability | $C_{41}$ | 0.0760 | 0.0802 | 0.0846 | 0.0911 | 0.0900 | 0.0845 |
| | $C_{42}$ | 0.0670 | 0.0656 | 0.0657 | 0.0646 | 0.0628 | 0.0621 |
| | $C_{43}$ | 0.0710 | 0.0688 | 0.0701 | 0.0790 | 0.0794 | 0.0774 |
| Electronic-commerce | $C_{51}$ | 0.0611 | 0.0664 | 0.0691 | 0.0656 | 0.0653 | 0.0617 |
| | $C_{52}$ | 0.0815 | 0.0802 | 0.0746 | 0.0731 | 0.0742 | 0.0723 |
| | $C_{53}$ | 0.0277 | 0.0265 | 0.0265 | 0.0261 | 0.0259 | 0.0255 |

**Table 4. The best and the worst indicator chosen by 5 experts.**

| Expert number | The best indicator | The worst indicator |
|---|---|---|
| 1 | The number of internet broadband access users ($C_{12}$) | Electronic-commerce purchase amount ($C_{52}$) |
| 2 | The number of websites per 100 enterprises owned ($C_{21}$) | The proportion of enterprises with electronic-commerce ($C_{53}$) |
| 3 | Research & development funds for industrial enterprises above designated size ($C_{42}$) | Express quantity ($C_{31}$) |
| 4 | Research & development funds for industrial enterprises above designated size ($C_{42}$) | The proportion of enterprises with electronic-commerce ($C_{53}$) |
| 5 | The proportion of enterprises with electronic-commerce ($C_{53}$) | Express quantity ($C_{31}$) |

demonstrates an equal important degree between the best indicator and others, while the value of 9 means the best one is extremely more critical and desirable than others. Values increasing from 1 to 9 indicate the crucial and desirable degree increase progressively. Results of the significance comparison for the best one with others are illustrated in Table 5 and results of the important degree comparison for others with the worst one are described in Table 6.

Thirdly, in terms of the comparison results above, subjective weights of all indicators are calculated employing a linear model shown in Eq (8). Results of subjective weights are demonstrated in Table 7, and afterwards an average value of weight for each indicator is calculated to attain the final subjective weight of each indicator. Results of the coherence indicator $\xi^*$ judged by 5 experts are also listed in Table 7. And the consistent ratio of 5 experts' judgment would be calculated in terms of Table 1 and Eq (9). It can be clearly found that the comparisons are of highly consistency as the consistent ratio are close to zero.

(3) Determine the integrated weights

Based on objective weights obtained via the AEW method from 2015 to 2020 and subjective weights obtained via the BWM, the integrated weights from 2015 to 2020 combining subjective and objective weights are obtained in accordance with the basic principle of moment estimation, which are computed by Eqs (10) and (11). The integrated weights regard of different years are shown in Table 8. The top four indicators with relatively greater integrated weights are $C_{42}$ with 0.0839 in 2020, $C_{31}$ representing express quantity with 0.0838 in 2020, $C_{23}$ representing express business income with 0.0746 in 2020, and $C_{41}$ with 0.0745 in 2020. And the last three indicators with relatively less integrated weights are $C_{13}$ representing long distance optical cable line length with 0.0406 in 2020, $C_{11}$ representing telephone penetration ratio with 0.0391 in 2020, and $C_{32}$ representing average wage of urban employees in information transmission, computer services and software industries with 0.0320 in 2020.

The comparison of integrated weights on criteria level in different years is depicted in Fig 3. It is illustrated that the integrated weights of $C_2$ on behalf of integrated development are all

**Table 5. Results of the significance comparison for the best sub-criterion with others.**

| Expert number | Best sub-criterion | $C_{11}$ | $C_{12}$ | $C_{13}$ | $C_{21}$ | $C_{22}$ | $C_{23}$ | $C_{24}$ | $C_{25}$ | $C_{31}$ | $C_{32}$ | $C_{33}$ | $C_{41}$ | $C_{42}$ | $C_{43}$ | $C_{51}$ | $C_{52}$ | $C_{53}$ |
|---|---|---|---|---|---|---|---|---|---|---|---|---|---|---|---|---|---|---|
| 1 | $C_{12}$ | 2 | 1 | 2 | 3 | 4 | 4 | 4 | 4 | 7 | 6 | 7 | 5 | 5 | 5 | 8 | 9 | 8 |
| 2 | $C_{21}$ | 5 | 4 | 5 | 1 | 2 | 2 | 2 | 2 | 6 | 6 | 6 | 3 | 3 | 3 | 7 | 7 | 8 |
| 3 | $C_{42}$ | 5 | 5 | 5 | 3 | 4 | 4 | 4 | 4 | 8 | 7 | 7 | 2 | 1 | 2 | 6 | 6 | 6 |
| 4 | $C_{42}$ | 7 | 6 | 7 | 5 | 5 | 5 | 5 | 5 | 4 | 3 | 3 | 2 | 1 | 2 | 8 | 8 | 9 |
| 5 | $C_{53}$ | 7 | 6 | 7 | 5 | 5 | 5 | 5 | 5 | 9 | 8 | 8 | 4 | 3 | 4 | 2 | 2 | 1 |

**Table 6. Results of the significance comparison for others with the worst sub-criterion.**

| Expert number | 1 | 2 | 3 | 4 | 5 |
|---|---|---|---|---|---|
| The worst sub-criteria Sub-criterion | $C_{52}$ | $C_{53}$ | $C_{31}$ | $C_{53}$ | $C_{31}$ |
| $C_{11}$ | 8 | 5 | 5 | 3 | 3 |
| $C_{12}$ | 9 | 6 | 5 | 4 | 4 |
| $C_{13}$ | 8 | 5 | 5 | 3 | 3 |
| $C_{21}$ | 7 | 9 | 7 | 5 | 5 |
| $C_{22}$ | 6 | 8 | 6 | 5 | 5 |
| $C_{23}$ | 6 | 8 | 6 | 5 | 5 |
| $C_{24}$ | 6 | 8 | 6 | 5 | 5 |
| $C_{25}$ | 6 | 8 | 6 | 5 | 5 |
| $C_{31}$ | 3 | 4 | 1 | 6 | 1 |
| $C_{32}$ | 4 | 4 | 3 | 7 | 2 |
| $C_{33}$ | 3 | 4 | 3 | 7 | 2 |
| $C_{41}$ | 5 | 7 | 8 | 8 | 6 |
| $C_{42}$ | 5 | 7 | 9 | 9 | 7 |
| $C_{43}$ | 5 | 7 | 8 | 8 | 6 |
| $C_{51}$ | 2 | 3 | 4 | 2 | 8 |
| $C_{52}$ | 1 | 3 | 4 | 2 | 8 |
| $C_{53}$ | 2 | 1 | 4 | 1 | 9 |

largest in different years, implying that the integrated development of informatization and industrialization is of great significance in DE development. The integrated weights of $C_4$ representing innovation ability ranks the second, implying that the number of patent applications

**Table 7. Subjective weights and consistency ratio.**

| Sub-criteria | Expert 1 | Expert 2 | Expert 3 | Expert 4 | Expert 5 | Average weight |
|---|---|---|---|---|---|---|
| $C_{11}$ | 0.1104 | 0.0381 | 0.0435 | 0.0317 | 0.0323 | 0.0512 |
| $C_{12}$ | 0.1814 | 0.0953 | 0.0435 | 0.0369 | 0.0376 | 0.0789 |
| $C_{13}$ | 0.1104 | 0.0381 | 0.0435 | 0.0317 | 0.0323 | 0.0512 |
| $C_{21}$ | 0.0736 | 0.1429 | 0.0726 | 0.0443 | 0.0452 | 0.0757 |
| $C_{22}$ | 0.0552 | 0.0476 | 0.0544 | 0.0443 | 0.0452 | 0.0493 |
| $C_{23}$ | 0.0552 | 0.0953 | 0.0544 | 0.0443 | 0.0452 | 0.0589 |
| $C_{24}$ | 0.0552 | 0.0953 | 0.0544 | 0.0443 | 0.0452 | 0.0589 |
| $C_{25}$ | 0.0552 | 0.0953 | 0.0544 | 0.0443 | 0.0452 | 0.0589 |
| $C_{31}$ | 0.0315 | 0.0318 | 0.0163 | 0.0554 | 0.0161 | 0.0302 |
| $C_{32}$ | 0.0368 | 0.0318 | 0.0311 | 0.0739 | 0.0282 | 0.0404 |
| $C_{33}$ | 0.0315 | 0.0318 | 0.0311 | 0.0739 | 0.0282 | 0.0393 |
| $C_{41}$ | 0.0442 | 0.0635 | 0.1088 | 0.1108 | 0.0565 | 0.0768 |
| $C_{42}$ | 0.0442 | 0.0635 | 0.1741 | 0.1821 | 0.0753 | 0.1078 |
| $C_{43}$ | 0.0442 | 0.0635 | 0.1088 | 0.1108 | 0.0565 | 0.0768 |
| $C_{51}$ | 0.0276 | 0.0272 | 0.0363 | 0.0277 | 0.1129 | 0.0463 |
| $C_{52}$ | 0.0158 | 0.0272 | 0.0363 | 0.0277 | 0.1129 | 0.0440 |
| $C_{53}$ | 0.0276 | 0.0119 | 0.0363 | 0.0158 | 0.1855 | 0.0554 |
| $\xi^*$ | 0.0394 | 0.0476 | 0.0435 | 0.0396 | 0.0403 | |
| Consistency ratio | 0.0075 | 0.0106 | 0.0097 | 0.0076 | 0.0077 | |

**Table 8. Integrated weights for all indicators regard of different years.**

| Sub-criteria | 2015 | 2016 | 2017 | 2018 | 2019 | 2020 |
|---|---|---|---|---|---|---|
| $C_{11}$ | 0.0398 | 0.0396 | 0.0395 | 0.0394 | 0.0393 | 0.0391 |
| $C_{12}$ | 0.0613 | 0.0611 | 0.0609 | 0.0606 | 0.0604 | 0.0602 |
| $C_{13}$ | 0.0407 | 0.0405 | 0.0403 | 0.0402 | 0.0408 | 0.0406 |
| $C_{21}$ | 0.0587 | 0.0587 | 0.0586 | 0.0586 | 0.0584 | 0.0583 |
| $C_{22}$ | 0.0647 | 0.0646 | 0.0640 | 0.0624 | 0.0610 | 0.0648 |
| $C_{23}$ | 0.0765 | 0.0748 | 0.0750 | 0.0726 | 0.0746 | 0.0746 |
| $C_{24}$ | 0.0499 | 0.0517 | 0.0499 | 0.0477 | 0.0470 | 0.0468 |
| $C_{25}$ | 0.0625 | 0.0635 | 0.0636 | 0.0663 | 0.0655 | 0.0692 |
| $C_{31}$ | 0.0808 | 0.0782 | 0.0803 | 0.0792 | 0.0830 | 0.0838 |
| $C_{32}$ | 0.0327 | 0.0325 | 0.0323 | 0.0322 | 0.0321 | 0.0320 |
| $C_{33}$ | 0.0492 | 0.0488 | 0.0501 | 0.0521 | 0.0504 | 0.0512 |
| $C_{41}$ | 0.0711 | 0.0730 | 0.0750 | 0.0783 | 0.0775 | 0.0745 |
| $C_{42}$ | 0.0858 | 0.0854 | 0.0852 | 0.0848 | 0.0843 | 0.0839 |
| $C_{43}$ | 0.0689 | 0.0679 | 0.0682 | 0.0721 | 0.0721 | 0.0710 |
| $C_{51}$ | 0.0509 | 0.0540 | 0.0556 | 0.0533 | 0.0530 | 0.0508 |
| $C_{52}$ | 0.0636 | 0.0627 | 0.0586 | 0.0575 | 0.0581 | 0.0567 |
| $C_{53}$ | 0.0430 | 0.0428 | 0.0427 | 0.0426 | 0.0425 | 0.0424 |

of industrial enterprises above designated size, research & development funds for industrial enterprises above designated size, and the number of research & development personnel in enterprises above designated size make great contributions to DE development. Considering that the economic development level of various regions in China is high and the infrastructure is comparatively perfect at present, the integrated weights of $C_1$ on behalf of digital infrastructure in different years are all the smallest. The integrated weights of $C_3$ representing social benefits are relatively close to those of $C_5$ representing electronic-commerce from 2015 to 2017. However, recently, with the development of the electronic-commerce industry becoming

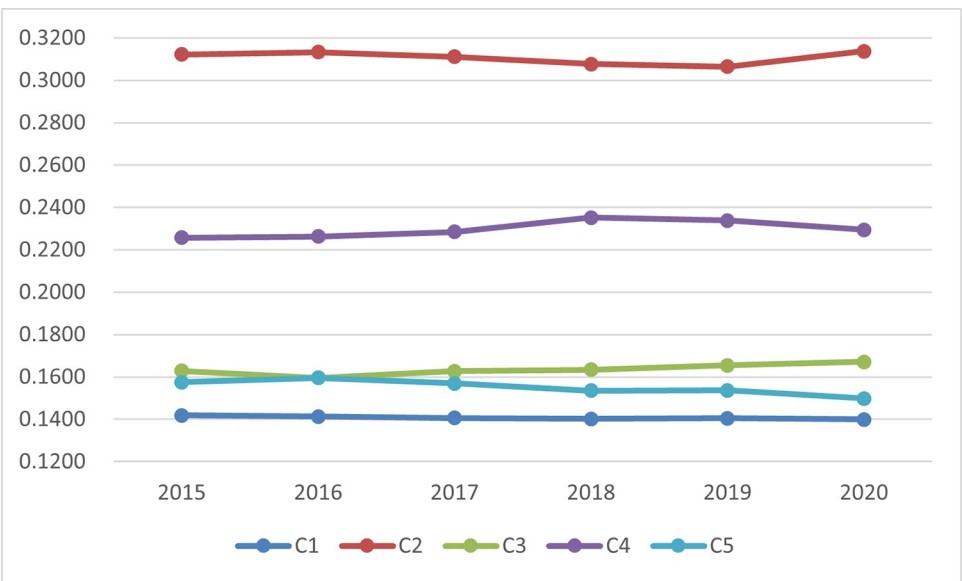

**Fig 3. Comparison of integrated weights on criteria level in different years.**

increasingly mature, the significance of social benefits in the development evaluation of DE is becoming increasingly prominent.

The difference of objective weights obtained via the AEW method from the year of 2015 to 2020 and subjective weights obtained via the BWM on indicators level are illustrated in Fig 4. We can find that there exists little difference in the objective weights of various years calculated utilizing actual data using the AEW method, while subjective weights obtained based on experts' opinions and judgments using the BWM are quite different from the objective weights of various years. Taking various weights of $C_{42}$ as an example, the subjective weight of $C_{42}$ is 0.1078 which is the greatest subjective weights, while the objective weights from 2015 to 2020 are 0.0670, 0.0656, 0.0657, 0.0646, 0.0628, and 0.0621, respectively, of which the objective weights are at the midstream level among all objective weights computed. Thus, the above comparison demonstrates that there exist significant differences between objective weights and subjective weights for the same indicator. Therefore, integrated weights combing subjective and objective weights should be calculated to assess the development of DE to avoid the bias in evaluation results via utilizing single subjective weights or objective weights.

## 5.2 Results of DE development evaluation for 31 provincial level regions in China

According to the objective data of indicators with regard to 31 regions in China from 2015 to 2020, the normalization matrix can be obtained via Eqs (15) and (16). Through combining with the integrated weights, the weighted normalization decision matrix is attained via the Eq (17). Afterwards, the utility function values $f(K_j)$ of 31 provincial level regions can be calculated based on Eqs (18)–(23). The utility function values of 31 regions from 2015 to 2020 are illustrated in Fig 5.

As is demonstrated in Fig 5, the values of utility function in Guangdong, Jiangsu, Zhejiang, Beijing, Shandong, Shanghai, and Sichuan are much greater than other provincial level regions, which are all greater than 0.30 from 2015 to 2020. Among them, Guangdong, Jiangsu,

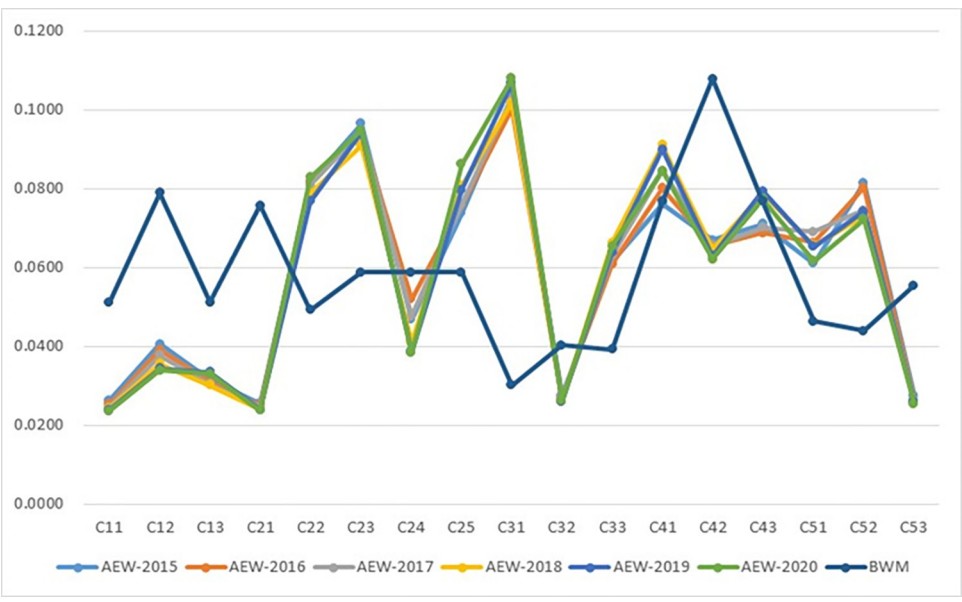

**Fig 4. The difference of objective weights obtained via the AEW method from 2015 to 2020 and subjective weights obtained via the BWM.**

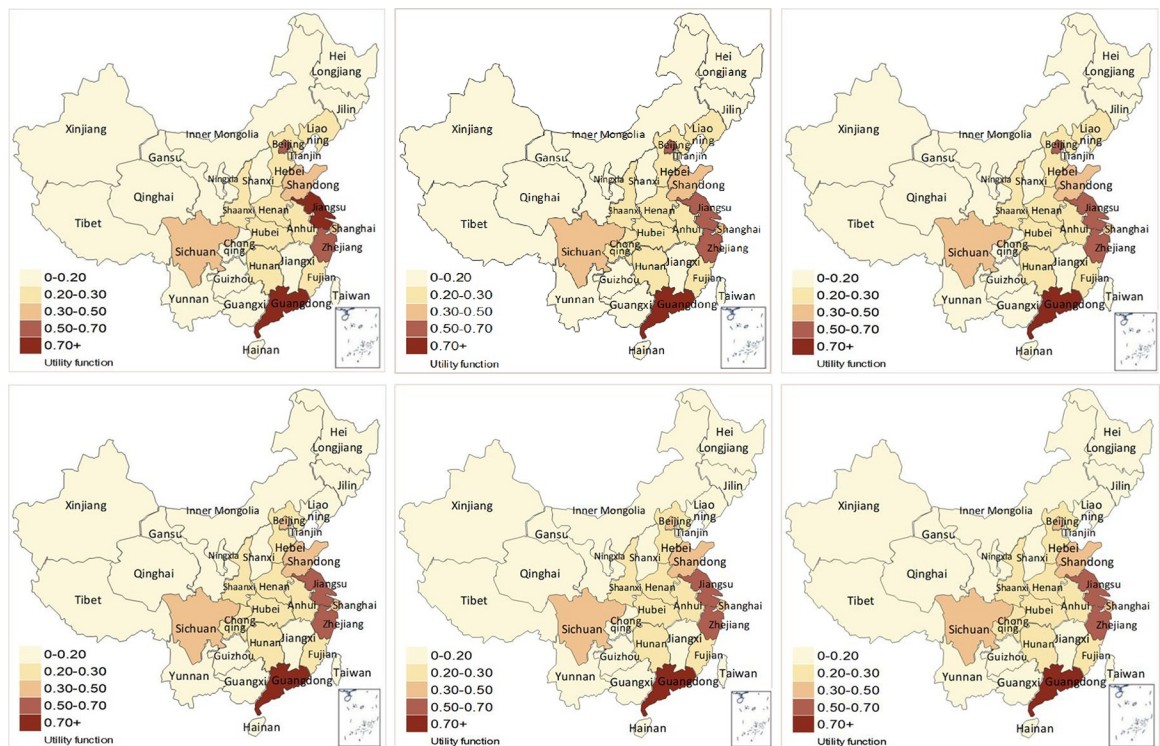

**Fig 5. The utility function values of 31 provincial level regions from 2015 to 2020.** (Reprinted from [41] under a CC BY license, with permission from Professor Sen Guo, original copyright 2019). (a) The utility function values of 31 regions in 2015. (b) The utility function values of 31 regions in 2016. (c) The utility function values of 31 regions in 2017. (d) The utility function values of 31 regions in 2018. (e) The utility function values of 31 regions in 2019. (f) The utility function values of 31 regions in 2020.

Zhejiang, and Beijing always maintain in the top four. Guangdong province has conscientiously implemented the planning requirements of "the implementation plan of the national pilot zone for innovative development of DE", and Shenzhen, known as the scientific and technological innovation center located in Guangdong province, has congregated a large amount of high-tech enterprises, communication enterprises, and software enterprises which can greatly accelerate the development of DE, hence, software business income, income from information technology services, employment of urban units in information transmission, software and information technology services, and the number of patent applications and research & development funds of industrial enterprises above designated size are much higher than other regions. Therefore, the development level of DE in Guangdong takes the lead from 2015 to 2020 with the average value of utility function higher than 0.90.

As an experimental field for building a smart digital city, the eastern coastal areas make sufficient use of the talent, technology, infrastructure and other resources brought by economically developed areas to vigorously carry out digital construction. Hence, the development level of DE of Jiangsu and Zhejiang ranked the second and the third, respectively, of which the average values of utility functions from 2015 to 2020 are 0.6398 and 0.5596, respectively. As the political and economic center, Beijing attaches importance to the investment of knowledge-based talents and owns high telephone penetration ratio, software business income, information technology service income, and relatively higher proportion of enterprises with electronic-commerce transactions. Beijing also highlights the appliance of big data, cloud computing, AI, and other techniques in decision-making and economic activities, which can

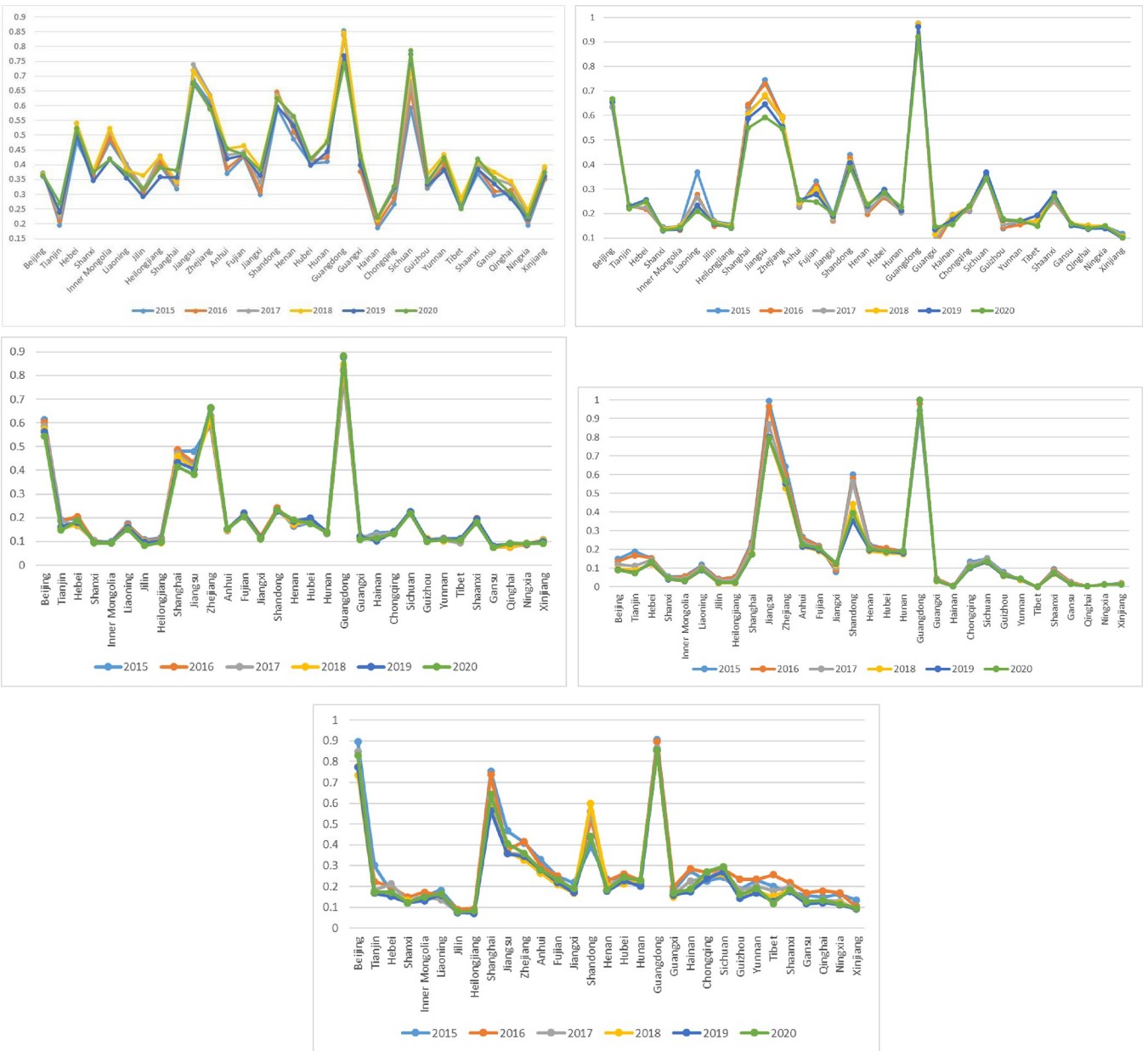

**Fig 6. The detailed performance of 31 regions from 2015 to 2020.** (a) Digital infrastructure performance of 31 provincial level regions from 2015 to 2020. (b) Integrated development performance of 31 regions from 2015 to 2020. (c) Social benefits performance of 31 regions from 2015 to 2020. (d) Innovation ability performance of 31 regions from 2015 to 2020. (e) Electronic-commerce performance of 31 regions from 2015 to 2020.

accelerate the development of DE to a great extent. Hence, Beijing ranked the fourth, of which the average value of utility function from 2015 to 2020 is 0.5009.

As a major industrial province, Shandong lays stress on the development of industrial internet and speeds up the integrated development of information technologies and traditional economy. According to the values of utility function calculated by MARCOS, in 2017 and 2018, the development level of DE in Shandong surpassed that in Shanghai, ranking the fifth. However, the telephone penetration ratio, the number of websites per 100 enterprises owned, average wage of urban employees in information transmission, computer services and software

industries, electronic-commerce sales amount, and the proportion of enterprises with electronic-commerce transactions in Shanghai are much higher than Shandong in 2019 and 2020, so that Shanghai surpassed Shandong ranked the fifth in 2019 and 2020.

Due to the late economic development in the southwest and northwest regions in China, the relevant talents, technology and basic facilities are still accumulated and constructed, and so that the number of internet broadband access users, the employment of urban units in information transmission, software and information technology services, the number of patent applications and research & development funds of industrial enterprises above designated size, and the number of research & development personnel in enterprises above designated size are much lower than other regions. Thus, the development level of the DE is obviously lower than others. Among them, Xinjiang and Ningxia ranked at the last two positions, of which the average values of utility function from 2015 to 2020 are 0.1218 and 0.1154, respectively. The development level of DE in the northeast region (especially Jilin and Heilongjiang provinces) rank at the downstream position of China. The primary causes are that the economy growth in the northeast region primarily relies on the heavy old industry and the primary industry and the industrial structure is relatively backward and the technology is relatively low. Hence, the number of websites per 100 enterprises owned, software business income, income from information technology services, research & development funds for industrial enterprises above designated size, and the number of research & development personnel in enterprises above designated size are relatively lower than other regions, so that the DE development level of these regions are inferior to others.

Fig 6 illustrates the detailed performance of 31 regions from 2015 to 2020, including digital infrastructure, integrated development, social benefits, innovation ability, and electronic-commerce performances.

(1) For digital infrastructure performance, generally, Guangdong performs the best followed by Sichuan and Shandong. The development of digital infrastructure in Sichuan province demonstrates an upward tendency year by year, and the value of utility function of digital infrastructure of Sichuan surpassed that of Guangdong in 2019 ranking the first. Consistent with the overall ranking, the digital infrastructure performance of the southwest and northwest regions in China still scores near the bottom, especially Ningxia province. Through comparing the utility function values of digital infrastructure performance in different years in 31 provincial level regions, it can be obviously discovered that the construction of digital basic facilities in different provincial level regions has been strengthened year after year, however, the development level of digital basic facilities in some regions (especially Zhejiang and Guangdong) in 2020 has dropped slightly brought by the influences of the COVID-19 epidemic.

(2) For integrated development performance, the annual value of utility function of integrated development in Guangdong achieves 0.9639 from 2015 to 2020, which is much higher than Jiangsu (the annual value of that is 0.6796) and Beijing (the annual value of that is 0.6494) ranking at the second and the third, respectively. The integrated development level of Shanghai, Zhejiang, Shandong and Sichuan are superior than other regions, ranking the fourth, fifth, sixth, and seventh, respectively. Consistent with the overall ranking, the integrated development level of the southwest, northeast, and northwest regions in China still scores near the bottom, especially Xinjiang, Qinghai, and Heilongjiang provinces. Through comparing the utility function values of integrated development in different years in 31 provincial level regions, it can be apparently found that the utility function values of integrated development in most provincial level regions have little difference in various years, nevertheless, the utility function values of integrated development in Guangdong, Liaoning,

Shanghai, Jiangsu, Zhejiang, Fujian, and Shandong provinces show a downward tendency with each passing year.

(3) For social benefits performance, the top three values of utility function of social benefits are originated from Guangdong, Zhejiang, and Beijing, with annual values of utility function from 2015 to 2020 to be 0.8396, 0.6303, and 0.5807, respectively, followed by Shanghai and Jiangsu. The performance of social benefits of other regions are much inferior to these five provincial level regions, among which the northwest regions in China are still at the bottom, especially Gansu, Qinghai, and Ningxia provinces. Through comparing the utility function values of social benefits in different years in 31 provincial level regions, it can be clearly found that the utility function values of social benefits in most regions vary a little in different years, except for Beijing, Tianjin, Shanghai, and Jiangsu, which illustrate a decrease trend from 2015 to 2020, while Guangdong indicates an obvious upward tendency from 2015 to 2020.

(4) For innovation ability performance, the annual values of utility function of innovation ability in Guangdong and Jiangsu provinces from 2015 to 2020 are much higher than other regions, with 0.9873 and 0.8711, respectively. Particularly, values of utility function of innovation ability in Guangdong from 2017 to 2020 all achieve 1, which demonstrates the innovation ability represented by the number of patent applications of industrial enterprises above designated size, research & development funds for industrial enterprises above designated size, and the number of research & development personnel in enterprises above designated size in Guangdong is much more superior than other regions being at the national leading level. The innovation ability of Zhejiang and Shandong rank the third and fourth with the annual values of utility function of innovation ability from 2015 to 2020 to be 0.5757 and 0.4893, respectively. The innovation ability of Qinghai, Hainan, and Tibet rank the last three positions. Through comparing the utility function values of innovation ability in different years in 31 provincial level regions, it can be summarized that the innovation ability in most regions shows an upward tendency with each passing year due to the increase input in research & development funds for industrial enterprises above designated size and the number of research & development personnel in enterprises above designated size.

(5) For electronic-commerce performance, the top four values of utility function of electronic-commerce can be stemmed from Guangdong, Beijing, Shanghai, and Shandong, with annual values of utility function from 2015 to 2020 to be 0.8708, 0.8093, 0.6485, and 0.4928, respectively, followed by Jiangsu and Zhejiang. Different from the overall rankings, the electronic-commerce performance of Heilongjiang and Jilin rank at the bottom owing to the relatively low level of electronic-commerce sales amount as well as the proportion of enterprises with electronic-commerce transactions. Through comparing the utility function values of electronic-commerce performance in different years in 31 provincial level regions, it indicates that the utility function values of electronic-commerce performance in most provincial level regions illustrate a downward tendency brought by the reduction of the proportion of enterprises with electronic-commerce transactions, while those of Shandong, Hunan, Chongqing, Sichuan, and Shaanxi illustrate an obvious upward trend due to the substantial increase of electronic-commerce sales amount and electronic-commerce purchase amount.

Through comparing the detailed performance of 31 regions from 2015 to 2020, it is obviously found that the DE development level of various regions is primarily affected by the

integrated development performance, innovation ability performance, and social benefits performance. Therefore, the backward regions in DE development, especially Guangxi, Jilin, Tibet, Qinghai, Xinjiang, and Ningxia, should emphasize the development of software industry and information technology industry, increase the input of research & development funds for industrial enterprises and the average wage of urban employees in information transmission, computer services and software industries, and propose talent incentive measures to increase the employment of urban units in information transmission, software and information technology services and the number of research & development personnel in enterprises above designated size, so that the development level of DE of these regions can be greatly improved.

## 6 Discussion on the robustness of the established MCDM model

To empirically prove the effectiveness and validation of the MCDM model combining Fuzzy-Delphi, AEW, BWM and MARCOS, 4 hybrid MCDM models are chosen to be comparison models and three ranking similarity coefficients are utilized to analyze ranking similarity.

4 comparison models for analyzing ranking similarity are indicated in Table 9. The detailed evaluation process of traditional TOPSIS and VIKOR can be referred to Behzadian et al. [42] and Zhao et al. [43]. Through comparatively discussing the ranking similarity among Model 1, Model 2, and Model 3, the validation of the established integrating weighting method is proved, and by discussing the ranking similarity among Model 1, Model 4, and Model 5, the validation of the evaluation model MARCOS is proved.

The similarity rankings are evaluated by three similarity coefficients of rankings, naming Spearman's ranking correlation coefficient ($r_S$), weighted Spearman's ranking correlation coefficient ($r_W$), and WS similarity coefficient ($WS$) in terms of Sałabun et al. [44] and Kizielewicz et al. [45]. These coefficients are calculated utilizing Eqs (24)–(26).

$$r_S = 1 - \frac{6 \times \sum_{i=1}^{N} (x_i - y_i)^2}{N \times (N^2 - 1)} \tag{24}$$

$$r_W = 1 - \frac{6 \times \sum_{i=1}^{N} (x_i - y_i)^2 [(N - x_i + 1) + (N - y_i + 1)]}{N \times (N^3 + N^2 - N - 1)} \tag{25}$$

$$WS = 1 - \sum_{i=1}^{N} \left( 2^{-x_i} \frac{|x_i - y_i|}{\max\{|x_i - 1|, |x_i - N|\}} \right) \tag{26}$$

where $x_i$ and $y_i$ demonstrate the sequences of rankings, and $N$ is the number of evaluated provincial level regions in the rankings.

It is important to note that in actual MCDM studies, the coherence of the top alternative is of great importance for evaluating the robustness of the evaluation results, as the top one is on

Table 9. The comparative models for ranking similarity analysis.

| Models | Model description |
|---|---|
| Model 1 | The proposed hybrid MCDM model integrating Fuzzy-Delphi, AEW, BWM and MARCOS |
| Model 2 | The hybrid MCDM model integrating Fuzzy-Delphi, AEW and MARCOS |
| Model 3 | The hybrid MCDM model integrating Fuzzy-Delphi, BWM and MARCOS |
| Model 4 | The hybrid MCDM model integrating Fuzzy-Delphi, AEW, BWM, and traditional TOPSIS |
| Model 5 | The hybrid MCDM model integrating Fuzzy-Delphi, AEW, BWM, and traditional VIKOR |

Note: Technique for order preference by similarity to an ideal solution is the full name of TOPSIS model and Vlsekriterijumska Optimizacija I Kompromisno Resenje is the full name of VIKOR model.

**Table 10. Similarity coefficients between the rankings of the established model and 4 comparison models.**

| Performance | Models | Similarity coefficients | | |
|---|---|---|---|---|
| | | $r_S$ | $r_W$ | WS |
| Overall performance | Model 1 | 1.0000 | 1.0000 | 1.0000 |
| | Model 2 | 1.0000 | 1.0000 | 1.0000 |
| | Model 3 | 1.0000 | 1.0000 | 1.0000 |
| | Model 4 | 1.0000 | 1.0000 | 1.0000 |
| | Model 5 | 1.0000 | 1.0000 | 1.0000 |
| Digital infrastructure performance | Model 1 | 1.0000 | 1.0000 | 1.0000 |
| | Model 2 | 1.0000 | 1.0000 | 1.0000 |
| | Model 3 | 1.0000 | 1.0000 | 1.0000 |
| | Model 4 | 0.9837 | 0.9641 | 0.9957 |
| | Model 5 | 0.9734 | 0.9628 | 0.9912 |
| Integrated development performance | Model 1 | 1.0000 | 1.0000 | 1.0000 |
| | Model 2 | 1.0000 | 1.0000 | 1.0000 |
| | Model 3 | 1.0000 | 1.0000 | 1.0000 |
| | Model 4 | 1.0000 | 1.0000 | 1.0000 |
| | Model 5 | 1.0000 | 1.0000 | 1.0000 |
| Social benefits performance | Model 1 | 1.0000 | 1.0000 | 1.0000 |
| | Model 2 | 1.0000 | 1.0000 | 1.0000 |
| | Model 3 | 1.0000 | 1.0000 | 1.0000 |
| | Model 4 | 0.9637 | 0.9542 | 0.9859 |
| | Model 5 | 0.9612 | 0.9533 | 0.9823 |
| Innovation ability performance | Model 1 | 1.0000 | 1.0000 | 1.0000 |
| | Model 2 | 1.0000 | 1.0000 | 1.0000 |
| | Model 3 | 1.0000 | 1.0000 | 1.0000 |
| | Model 4 | 1.0000 | 1.0000 | 1.0000 |
| | Model 5 | 1.0000 | 1.0000 | 1.0000 |
| Electronic-commerce performance | Model 1 | 1.0000 | 1.0000 | 1.0000 |
| | Model 2 | 1.0000 | 1.0000 | 1.0000 |
| | Model 3 | 1.0000 | 1.0000 | 1.0000 |
| | Model 4 | 0.9601 | 0.9531 | 0.9849 |
| | Model 5 | 0.9538 | 0.9501 | 0.9823 |

behalf of the optimal selection for actual decision making. Hence, the WS similarity coefficient (*WS*) makes the ranking variance of the top alternative exert much more influences on final ranking similarities.

In terms of the actual data of 17 sub-criteria in 31 provincial level regions in China, ranking results are calculated via 5 models from the overall performance and five sub-performances of DE development, including digital infrastructure performance, integrated development performance, social benefits performance, innovation ability performance, and electronic-commerce performance. Afterwards, similarity coefficients between ranking results of the established MCDM model and 4 comparison models are calculated via Eqs (24)-(26), listing in Table 10.

It can be discovered from Table 10 that:

(1) In terms of overall performance, integrated development performance, and innovation ability performance, rankings of Model 1 and 4 comparison models are highly coherent. All similarity coefficients are 1.000 meaning that after changing the weighting approach and

the comprehensive evaluation model, the ranking results of general performance, integrated development performance, and innovation ability performance have not altered, implying that the established MCDM methodology has superior robustness in estimating DE development.

(2) In terms of digital infrastructure performance, social benefits performance, and electronic-commerce performance, through comparing similarity coefficients of Models 1, 2, and 3, we can discover that all similarity coefficients of these three models are 1.0000 indicating that changing the weighting method will not influence ranking results, hence, the integrated weighting method employed in this paper has better robustness in evaluating DE development. Through comparing similarity coefficients of Models 4 and 5, it can be found that all similarity coefficients of these two models are all larger than 0.95, which demonstrates that a slight distinct exist in ranking results among Models 1, 4, and 5 in digital infrastructure, social benefits, and electronic-commerce performances. Specifically, similarity coefficients values of Model 4 are larger than those of Model 5, which illustrates that ranking results of Model 4 in digital infrastructure, social benefits, and electronic-commerce performances are relatively highly similar with those of Model 1. Moreover, values of $WS$ coefficient of Models 4 and 5 are all greater than 0.98, which means the top provincial level regions of Models 4 and 5 are highly homologous with those of Model 1 in terms of digital infrastructure, social benefits, and electronic-commerce performances. Therefore, it can be summarized that the change of comprehensive evaluation model will influence the ranking results of sub-performance to a certain degree, nevertheless the impacts are mild, that means the established MCDM model has good robustness in assessing sub-performance of DE development.

Therefore, the above ranking comparison discussion indicates that changing the weighting method exert no impact on ranking results of both overall performance and sub-performances, while changing comprehensive evaluation model will have slight impact on ranking results of sub-performances but will not affect ranking results of overall performance. Hence, the established MCDM methodology integrating Fuzzy-Delphi method, AEW method, BWM, and MARCOS model has superior robustness and validity in evaluating DE development.

## 7 Conclusions

With the deepening of a new round of scientific and technological revolution and industrial reform, digital technology with the internet as the core has been continuously innovated and widely penetrated into various economic fields, which has brought great changes in productivity and mode of production. The world has entered a new era that DE promotes economic development. In October 2020, China proposed to upgrade the DE into a national strategy, accelerate the development of DE, further accelerate digital industrialization and industrial digitization, deeply integrate the DE with the real economy, give sufficient play to the driving part of DE in economic growth as well as promote high-quality economic development. It is demonstrated that the DE has become the focus of China's economic development planning and a new engine to enhance national strength. Evaluating the development level of DE in various regions of China is conducive to timely find out the shortcomings in China's DE development, accurately grasp the key factors affecting China's economic growth, and then provide an important basis for putting forward corresponding policy suggestions to scientifically guide the direction of China's economic development. Therefore, coupled with the connotation and particular features of DE, this paper constructs the evaluation indicator system of DE development, and makes a comprehensive evaluation of the development level of DE in various regions of China.

Firstly, the initial indicator system is built from digital infrastructure, integrated development, social benefits, innovation ability, and electronic-commerce dimensions containing 31 initial sub-criteria through referring to related reports and published literature. Then, the Fuzzy-Delphi approach is applied to screen the crucial indicators considering 5 experts judgments. And the final index system is constructed containing 17 quantitative sub-criteria. The development level of DE in 31 regions in China will be evaluated. Aiming at comparatively analyzing the development of DE in different regions, actual data of 17 sub-criteria with regard to these 31 provincial level regions from 2015 to 2020 can be gathered from the website of National Bureau of Statistics. Afterwards, objective weights from 2015 to 2020 can be calculated via the AEW method based on actual data of 17 sub-criteria with regard to 31 provincial level regions from 2015 to 2020. The BWM is also applied to identify the subjective weights of 17 indicators in terms of experts' judgments on important degree of indicators. Then the integrated weights from 2015 to 2020 are computed via the basic principle of moment estimation in accordance with subjective weights determined through the BWM and the objective weights estimated via the AEW approach. The top four sub-criteria with relatively greater integrated weights are $C_{42}$ representing research & development funds for industrial enterprises above designated size with 0.0839 in 2020, $C_{31}$ representing express quantity with 0.0838 in 2020, $C_{23}$ representing express business income with 0.0746 in 2020, and $C_{41}$ representing the number of patent applications of industrial enterprises above designated size with 0.0745 in 2020. Moreover, the last three sub-criteria with relatively less integrated weights are $C_{13}$ representing long distance optical cable line length with 0.0406 in 2020, $C_{11}$ representing telephone penetration ratio with 0.0391 in 2020, and $C_{32}$ representing average wage of urban employees in information transmission, computer services and software industries with 0.0320 in 2020. From sub-performance perspectives, the integrated weights of $C_2$ on behalf of integrated development are all largest in different years, followed by $C_4$ representing innovation ability. While the integrated weights of $C_1$ on behalf of digital infrastructure in different years are all the smallest.

After determining the final indicator system for DE development evaluation and integrated weights of all sub-criteria, MARCOS model is applied to rank the development level of DE of 31 provincial level regions in China. Case analysis illustrates that the values of utility function in Guangdong, Jiangsu, Zhejiang, Beijing, Shandong, Shanghai, and Sichuan are much greater than other regions, among which, Guangdong, Jiangsu, Zhejiang, and Beijing always maintain in the top four from 2015 to 2020. Due to the late economic development in the southwest and northwest regions in China, the development level of the DE is obviously lower than that of other regions, among which, Xinjiang and Ningxia ranked at the last two positions, of which the average values of utility function are 0.1218 and 0.1154 from 2015 to 2020, respectively. The development level of DE in the northeast region (especially Jilin and Heilongjiang provinces) also rank at the downstream position of China. For the detailed performance of 31 regions from 2015 to 2020, Guangdong always has the best performance in digital infrastructure, integrated development, social benefits, innovation ability, and electronic-commerce performances, while the northwest regions in China are inferior to others in sub-performances. Through comparing the detailed performance of 31 regions from 2015 to 2020, we can summarize that the DE development level of various regions is primarily affected by the integrated development performance, innovation ability performance, and social benefits performance. Therefore, the backward regions in DE development, especially Guangxi, Jilin, Tibet, Qinghai, Xinjiang, and Ningxia, should emphasize the development of software industry and information technology industry, increase the input of research & development funds for industrial enterprises above designated size and the average wage of urban employees in information transmission, computer services and software industries, and propose talent incentive measures to increase the employment of urban units in information transmission, software and

information technology services and the number of research & development personnel in enterprises above designated size, so that the development level of DE of these regions can be greatly improved.

The robustness of the constructed MCDM model coupling Fuzzy-Delphi, AEW, BWM and MARCOS is discussed through selecting four comparison models and utilizing three similarity coefficients of rankings including $r_S$, $r_W$, and *WS* to estimate the ranking similarity. The ranking comparison discussion indicates that changing the weighting method will not change ranking results of both overall performance and sub-performance, while changing comprehensive evaluation model will have slight impact on ranking results of sub-performance but will not affect ranking results of overall performance. Hence, the constructed MCDM methodology coupling Fuzzy-Delphi method, AEW method, BWM, and MARCOS model has superior robustness and validity in evaluating DE development.

It needs to be noted that the hybrid MCDM model established in this investigation has wide applications. It cannot only be limited to estimate the DE development level in various regions of China, but also can be applied to other countries. However, there are still some limitations of this model. If there is only one region to be estimated, the AEW approach will be unapplicable to calculate the objective weights. For one evaluated alternative, the weight determination method with variance-driven principle, entropy weighting approach, and AEW method are all not applicable [46, 47]. And the MARCOS model is also inappropriate except for setting the positive and negative ideal solutions ahead of time. For one evaluated alternative issue, fuzzy theory-based evaluation method and matter element extension method are applicable for composing the MCDM model. Therefore, in the future research, it is necessary to build a comprehensive evaluation model for assessing the development level of DE suitable for different situations, and explore the driving factors of high-quality development of DE, so as to provide policy reference for high-quality development of DE.

## Supporting information

**S1 Data. Data of 17 sub-criteria in 31 regions of China from 2015 to 2020 are listed in the attached file named data.**
(XLSX)

## Author Contributions

**Conceptualization:** Haoran Zhao.

**Data curation:** Yuchen Wang.

**Formal analysis:** Haoran Zhao.

**Investigation:** Haoran Zhao.

**Methodology:** Haoran Zhao.

**Supervision:** Sen Guo.

**Writing – original draft:** Haoran Zhao, Yuchen Wang.

**Writing – review & editing:** Haoran Zhao, Sen Guo.

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
