## [Decision Letter · Decision Letter 0]

1 Feb 2023

PONE-D-22-33442A hybrid MCDM model combining Fuzzy-Delphi, AEW, BWM, and MARCOS for digital economy development comprehensive evaluation of 31 provincial level regions in ChinaPLOS ONE

Dear Dr. Zhao,

Thank you for submitting your manuscript to PLOS ONE. After careful consideration, we feel that it has merit but does not fully meet PLOS ONE’s publication criteria as it currently stands. Therefore, we invite you to submit a revised version of the manuscript that addresses the points raised during the review process.

We look forward to receiving your revised manuscript.

Kind regards,

Dragan Pamucar

Academic Editor

PLOS ONE

Journal Requirements:

"This paper is support by Qin Xin Talents Cultivation Program, Beijing Information Science & Technology University, under Grant No. QXTCPC202113."

"This paper is support by Qin Xin Talents Cultivation Program, Beijing Information Science & Technology University, under Grant No. QXTCPC202113."

5. We note that Figure 5 in your submission contain [map/satellite] images which may be copyrighted. All PLOS content is published under the Creative Commons Attribution License (CC BY 4.0), which means that the manuscript, images, and Supporting Information files will be freely available online, and any third party is permitted to access, download, copy, distribute, and use these materials in any way, even commercially, with proper attribution. For these reasons, we cannot publish previously copyrighted maps or satellite images created using proprietary data, such as Google software (Google Maps, Street View, and Earth). For more information, see our copyright guidelines: http://journals.plos.org/plosone/s/licenses-and-copyright.

a) You may seek permission from the original copyright holder of Figure 5 to publish the content specifically under the CC BY 4.0 license.  

Reviewers' comments:

Reviewer's Responses to Questions

**Comments to the Author**

1. Is the manuscript technically sound, and do the data support the conclusions?

Reviewer #1: Yes

Reviewer #2: Yes

2. Has the statistical analysis been performed appropriately and rigorously? 

Reviewer #1: Yes

Reviewer #2: Yes

3. Have the authors made all data underlying the findings in their manuscript fully available?

Reviewer #1: Yes

Reviewer #2: Yes

4. Is the manuscript presented in an intelligible fashion and written in standard English?

Reviewer #1: Yes

Reviewer #2: Yes

5. Review Comments to the Author

Reviewer #1: Manuscript title: A hybrid MCDM model combining Fuzzy-Delphi, AEW, BWM, and MARCOS for digital economy development comprehensive evaluation of 31 provincial level regions in China.

The manuscript is well organized and the contents fit with the journal’s topics. The methodology is well described and applied. I give recognition to the authors for their very high-quality work.

I suggest some minor paper corrections:

. The authors used modern methods in their work. I think it would be good to expand the literature analysis and show the application of these methods (AEW, BWM, MARCOS) in some other research, such as: Bakir, M., Akan, Ş., Özdemir, E. (2021) regional aircraft selection with fuzzy PIPRECIA and fuzzy MARCOS: A case study of the Turkish airline industry. Facta Universitatis, Series: Mechanical Engineering, 19(3), 423-445, doi:https://doi.org/10.22190/FUME210505053B; Torğul, B., Demiralay, E., & Paksoy, T. (2022). Training aircraft selection for department of flight training in fuzzy environment. Decision Making: Applications in Management and Engineering, 5(1), 264-289. https://doi.org/10.31181/dmame0311022022t; Fazlollahtabar, H., Kazemitash, N. (2021). Green supplier selection based on the information system performance evaluation using the integrated best-worst method. Facta Universitatis, Series: Mechanical Engineering, 19(3), 345-360. doi:https://doi.org/10.22190/FUME201125029F. In this way, the authors would confirm the quality of the methods they used in their work.

. I think the best place for figure 1 is between the title of section 2 and the subtitle of section 2.1. Of course, this figure should be accompanied by at least one paragraph of text explaining the figure and announcing the description of the methods.

. In conclusion, several clear directions for further research should be given.

Reviewer #2: The paper A hybrid MCDM model combining Fuzzy-Delphi, AEW, BWM, and MARCOS for digital economy development comprehensive evaluation of 31 provincial level regions in China falls within the scope of the journal PONE and represented a very interesting study with strong and recently developed approaches.

Structure of the paper is well with clear explanations. The authors have made integration of different approaches into one unique model. Definitely, the paper deserves attention due to having great potential to be published.

The paper should be improved according to following suggestions:

- Literature review should be a separate section.

- The follwoing reference should be added:

1) Stević, Ž., Subotić, M., Softić, E., & Božić, B. (2022). Multi-Criteria Decision-Making Model for Evaluating Safety of Road Sections, J. Intell. Manag. Decis., 1(2), 78-87. https://doi.org/10.56578/jimd010201

2) Kar, B., Mohapatra, B., Kar, S., & Tripathy, S. (2022). Small and Medium Enterprise Debt Decision: A Best-Worst Method Framework. Operational Research in Engineering Sciences: Theory and Applications.

3) Bakır, M., & Atalık, Ö. (2021). Application of fuzzy AHP and fuzzy MARCOS approach for the evaluation of e-service quality in the airline industry. Decision Making: Applications in Management and Engineering, 4(1), 127-152.

- Error! Reference source not found. Page 11 should be corrected.

- Explain reasons for the integration of these approaches in a unique model.

- Table 2 should be smaller. Now is out of margins.

6. PLOS authors have the option to publish the peer review history of their article (what does this mean?). If published, this will include your full peer review and any attached files.

Reviewer #1: No

Reviewer #2: No

---

## [Author Response · Author response to Decision Letter 0]

6 Mar 2023

Response to reviewers

Dear Editor,

Thank you very much for your work. Thanks a lot for the reviewers’ comments， careful check, and their kind suggestions on our manuscript. We provide this cover letter to explain, point by point, the details of our revisions in the manuscript and our responses to the reviewers’ comments as follows. In order to make the changes easily viewable for you and reviewers, we marked the revisions in the revised manuscript in red color. We hope the revised manuscript would satisfy you and reviewers. We are looking forward to hearing from you soon. 

Best regards,

Haoran Zhao

1 Response to Reviewer 1

Revisions list according to the suggestions from Reviewer 1:

Manuscript title: A hybrid MCDM model combining Fuzzy-Delphi, AEW, BWM, and MARCOS for digital economy development comprehensive evaluation of 31 provincial level regions in China. The manuscript is well organized and the contents fit with the journal’s topics. The methodology is well described and applied. I give recognition to the authors for their very high-quality work.

I suggest some minor paper corrections:

1. The authors used modern methods in their work. I think it would be good to expand the literature analysis and show the application of these methods (AEW, BWM, MARCOS) in some other research, such as: Bakir, M., Akan, Ş., Özdemir, E. (2021) regional aircraft selection with fuzzy PIPRECIA and fuzzy MARCOS: A case study of the Turkish airline industry. Facta Universitatis, Series: Mechanical Engineering, 19(3), 423-445, doi:https://doi.org/10.22190/FUME210505053B; Torğul, B., Demiralay, E., & Paksoy, T. (2022). Training aircraft selection for department of flight training in fuzzy environment. Decision Making: Applications in Management and Engineering, 5(1), 264-289. https://doi.org/10.31181/dmame0311022022t; Fazlollahtabar, H., Kazemitash, N. (2021). Green supplier selection based on the information system performance evaluation using the integrated best-worst method. Facta Universitatis, Series: Mechanical Engineering, 19(3), 345-360. doi:https://doi.org/10.22190/FUME201125029F. In this way, the authors would confirm the quality of the methods they used in their work.

Thank you very much for your comment and suggestion. We have added these literatures in the manuscript. Please check the content marked in red color in Literature review section and Reference section.

2. I think the best place for figure 1 is between the title of section 2 and the subtitle of section 2.1. Of course, this figure should be accompanied by at least one paragraph of text explaining the figure and announcing the description of the methods.

Thank you very much for your comment and suggestion. We have changed the place of Figure 1 between the title of Section 3 and the subtitle of Section 3.1. And a paragraph is added to announce the description of the methods. The concrete description of the MCDM framework is introduced in Section 3.4. Please check revisions in Section 3 in the manuscript.

3. In conclusion, several clear directions for further research should be given.

Thank you very much for your comment and suggestion. We have added the directions for future research at the end of Conclusion section. Please check the added content marked in red color in the manuscript.

2 Response to Reviewer 2

Revisions list according to the suggestions from Reviewer 2:

The paper A hybrid MCDM model combining Fuzzy-Delphi, AEW, BWM, and MARCOS for digital economy development comprehensive evaluation of 31 provincial level regions in China falls within the scope of the journal PONE and represented a very interesting study with strong and recently developed approaches.

Structure of the paper is well with clear explanations. The authors have made integration of different approaches into one unique model. Definitely, the paper deserves attention due to having great potential to be published.

The paper should be improved according to following suggestions:

1. Literature review should be a separate section.

Thank you very much for your comment and suggestion. We have separated Literature review from Introduction section. Please check Section 2 in the manuscript.

2. The follwoing reference should be added:

1) Stević, Ž., Subotić, M., Softić, E., & Božić, B. (2022). Multi-Criteria Decision-Making Model for Evaluating Safety of Road Sections, J. Intell. Manag. Decis., 1(2), 78-87. https://doi.org/10.56578/jimd010201

2) Kar, B., Mohapatra, B., Kar, S., & Tripathy, S. (2022). Small and Medium Enterprise Debt Decision: A Best-Worst Method Framework. Operational Research in Engineering Sciences: Theory and Applications.

3) Bakır, M., & Atalık, Ö. (2021). Application of fuzzy AHP and fuzzy MARCOS approach for the evaluation of e-service quality in the airline industry. Decision Making: Applications in Management and Engineering, 4(1), 127-152.

Thank you very much for your comment and suggestion. We have added these literatures in the manuscript. Please check the content marked in red color in Literature review section and Reference section.

3. Error! Reference source not found. Page 11 should be corrected.

Thank you very much for your comment and suggestion. We have corrected the reference source in the manuscript. Please check the revisions in the manuscript.

4. Explain reasons for the integration of these approaches in a unique model.

Thank you very much for your comment. We have added the explanation for the integration of these approaches in a unique model in Literature review section. Please check the contents marked in red color in the manuscript. The index system for evaluating the DE development is established based on Fuzzy-Delphi considering about experts knowledge and experience. And the BWM is employed to determine the subjective weights, as it only needs to compare the importance of each sub-criterion with the best and the worst sub-criteria, which is time-saving and convenient based on experts’ judgments. And the AEW method is utilized to determine the objective weights based on objective data. These two methods have been employed in many fields to determine sub-criteria weights in comprehensive evaluation. To comprehensively considering the importance of experts judgments and objective data, a weight integrating method is employed to combining subjective weights and objective weights based on the basic principle of moments estimation. Then integrated weights can be obtained. Afterwards, a new MCDM method named MARCOS, proposed by Željko Steviā and Dragan Pamučar in 2020, will be employed to estimate the development level of DE in 22 provinces, 5 autonomous areas, and 4 provincial level megacities in China. MARCOS model can take the positive and negative ideal solutions into account at the same time, and rank the provincial level regions based on the utility functions, which can make the results have superior robustness and accuracy. Therefore, a MCDM framework combining Fuzzy-Delphi, the BWM, the AEW, and MARCOS methods is established for DE development comprehensive evaluation of 31 provincial level regions in China.

5. Table 2 should be smaller. Now is out of margins.

Thank you very much for your comment and suggestion. We have made Table 2 become smaller. Please check the revisions in Table 2.

3 Response to Editor

Thank you very much for your comment and suggestion. We have revised our manuscript’s style to meet PLOS ONE's style requirements.

Thank you very much for your comment. Our Funding Statement is: 

"This paper is support by Qin Xin Talents Cultivation Program, Beijing Information Science & Technology University, under Grant No. QXTCPC202113."

And when we resubmit, we will provide the correct grant numbers for the awards we received for this study in the‘Funding Information’ section.

"This paper is support by Qin Xin Talents Cultivation Program, Beijing Information Science & Technology University, under Grant No. QXTCPC202113."

"This paper is support by Qin Xin Talents Cultivation Program, Beijing Information Science & Technology University, under Grant No. QXTCPC202113."

Thank you very much for your comment. We have removed funding-related text from the manuscript. And our Funding Statement is: 

"This paper is support by Qin Xin Talents Cultivation Program, Beijing Information Science & Technology University, under Grant No. QXTCPC202113."

Thank you very much for changing the online submission form.

Thank you very much for your comment and suggestion. But our research does not include human subjects research, animal research or global research and authors do not have competing interest. Hence, we think we do not need to attach ethics statement.

5. We note that Figure 5 in your submission contain [map/satellite] images which may be copyrighted. All PLOS content is published under the Creative Commons Attribution License (CC BY 4.0), which means that the manuscript, images, and Supporting Information files will be freely available online, and any third party is permitted to access, download, copy, distribute, and use these materials in any way, even commercially, with proper attribution. For these reasons, we cannot publish previously copyrighted maps or satellite images created using proprietary data, such as Google software (Google Maps, Street View, and Earth). For more information, see our copyright guidelines: http://journals.plos.org/plosone/s/licenses-and-copyright.

 a) You may seek permission from the original copyright holder of Figure 5 to publish the content specifically under the CC BY 4.0 license. 

Thank you very much for your comment. We have contacted the original copyright holder with the Content Permission Form and uploaded the completed Content Permission Form as an "Other" file with our submission. We have also added ‘Reprinted from [41] under a CC BY license, with permission from Professor Sen Guo, original copyright 2019.’ in the figure caption of Figure 5. Please check the added content.

Thank you very much for your comment. We have attached a file named Data in Supporting Information section to list data of 17 sub-criteria in 31 regions of China from 2015 to 2020. And we have included captions for the supporting information file at the end of our manuscript and cited it in Section 5. Please check the added content.

---

## [Decision Letter · Decision Letter 1]

13 Mar 2023

A hybrid MCDM model combining Fuzzy-Delphi, AEW, BWM, and MARCOS for digital economy development comprehensive evaluation of 31 provincial level regions in China

PONE-D-22-33442R1

Dear Dr. Zhao,

We’re pleased to inform you that your manuscript has been judged scientifically suitable for publication and will be formally accepted for publication once it meets all outstanding technical requirements.

Kind regards,

Dragan Pamucar

Academic Editor

PLOS ONE

Additional Editor Comments (optional):

Reviewers' comments:

Reviewer's Responses to Questions

**Comments to the Author**

1. If the authors have adequately addressed your comments raised in a previous round of review and you feel that this manuscript is now acceptable for publication, you may indicate that here to bypass the “Comments to the Author” section, enter your conflict of interest statement in the “Confidential to Editor” section, and submit your "Accept" recommendation.

Reviewer #1: All comments have been addressed

Reviewer #2: All comments have been addressed

2. Is the manuscript technically sound, and do the data support the conclusions?

Reviewer #1: Yes

Reviewer #2: Yes

3. Has the statistical analysis been performed appropriately and rigorously? 

Reviewer #1: Yes

Reviewer #2: Yes

4. Have the authors made all data underlying the findings in their manuscript fully available?

Reviewer #1: Yes

Reviewer #2: Yes

5. Is the manuscript presented in an intelligible fashion and written in standard English?

Reviewer #1: Yes

Reviewer #2: Yes

6. Review Comments to the Author

Reviewer #1: All the reviewers' comments have been addressed carefully and sufficiently. The revisions are rational from my point of view. I think the current version of the paper can be accepted.

Reviewer #2: The paper has been improved. Not of all my comments are adopted, but almost all, so my recommendation is to accept the paper.

7. PLOS authors have the option to publish the peer review history of their article (what does this mean?). If published, this will include your full peer review and any attached files.

Reviewer #1: No

Reviewer #2: No

---

## [Editor Report · Acceptance letter]

31 Mar 2023

PONE-D-22-33442R1 

A hybrid MCDM model combining Fuzzy-Delphi, AEW, BWM, and MARCOS for digital economy development comprehensive evaluation of 31 provincial level regions in China 

Dear Dr. Zhao:

I'm pleased to inform you that your manuscript has been deemed suitable for publication in PLOS ONE. Congratulations! Your manuscript is now with our production department. 

Kind regards, 

on behalf of

Dr. Dragan Pamucar 

Academic Editor

PLOS ONE